# Lattice-hydrogen cycling mechanism enables pH-universal hydrogen evolution at ampere-level current densities

Yan Zhang[1,2], Biao Feng[1,2], Jingyi Tian[1], Shiqi Zhou[1], Changkai Zhou[1], Yiqun Chen[1], Xiaoli Xia[1], Xizhang Wang[1], Lijun Yang [1], Luming Peng [1], Qiang Wu [1] ✉, Hongwen Huang [1] ✉ & Zheng Hu [1] ✉

Controllable supply of hydrogen intermediate across a wide pH range is crucial for electroreduction reactions, but is hindered by pH-dependent hydrogen species formation on conventional catalysts. We report a lattice-hydrogen cycling mechanism that dissociates hydrogen intermediate availability from electrolyte pH. By integrating proton-blocking Ru with thermally-hydrogenated $H_xWO_3$, we create a dynamic hydrogen reservoir, enabling efficient hydrogen supply. In-situ Raman spectroscopy, isotopic labeling, and theoretical simulations reveal the lattice hydrogen in $H_xWO_3$ migrates swiftly to Ru active sites via low-energy-barrier pathways, while consumed hydrogen is spontaneously replenished via proton adsorption (acidic) or water dissociation (alkaline/neutral). Consequently, this catalyst achieves a competitive pH-universal performance for hydrogen evolution reaction, with low overpotentials (125 mV acidic, 142 mV alkaline, 219 mV neutral @1 A cm$^{-2}$) alongside 500-hour stability.

The transition towards sustainable energy and chemical production necessitates innovative strategies to decouple human development from fossil fuel dependence[1]. Electrochemical hydrogen-involving reactions, such as the hydrogen evolution reaction (HER), $CO_2$ reduction ($CO_2$RR), and nitrogen reduction reaction (NRR), serve as critical bridges connecting renewable energy to sustainable fuel and chemical production[2,3]. In these processes, the kinetics of active hydrogen (H*) formation plays an important role in modulating the reaction pathways and reaction kinetics[2,4,5]. It is well established that the formation of H* is highly dependent on the pH of the electrolyte. In acidic media, the abundance of protons ($H^+$) facilitates the direct proton-electron coupling ($H^+ + e^- \rightarrow H^*$), while in neutral/alkaline conditions, water dissociation ($H_2O + e^- \rightarrow H^* + OH^-$) becomes a necessary step, introducing an additional energy barrier[6]. This fundamental dichotomy leads to the severe kinetic penalties in neutral/alkaline environments. As an example, the 2–3 order-of-magnitude drop in HER activity for Pt is observed when transitioning from acidic to alkaline electrolytes[7]. In

addition, for renewable electricity-driven electrochemical hydrogen-involving reactions (HER, $CO_2$RR, NRR, etc.), the intermittent and fluctuating electricity output induces variations in the current density of practical electrochemical devices, which in turn causes the pH oscillations at the electrified interface under operating conditions[8]. In this context, enabling the controllable and efficient supply of H* in all-pH electrolytes to overcome pH-imposed limitations is of both fundamental and industrial significance.

Focusing on the HER, the local pH value on the catalyst surface often experiences significant changes, especially in direct seawater electrolysis. For instance, the local pH variations exceeding 2 pH units were detected even in strong buffer electrolytes at a moderate current density of −30 mA cm$^{-2}$. The fluctuations will become more pronounced at ampere-level current density, which greatly affect HER performance[9]. One conventional strategy for designing pH-universal catalysts is to integrate an oxyphilic component with a catalytically active component, where the oxyphilic component typically enhances

[1]State Key Laboratory of Coordination Chemistry, Key Laboratory of Mesoscopic Chemistry of MOE, School of Chemistry and Chemical Engineering, Nanjing University, Nanjing, Jiangsu 210023, P. R. China. [2]These authors contributed equally: Yan Zhang, Biao Feng. ✉e-mail: wqchem@nju.edu.cn; huanghw@nju.edu.cn; zhenghu@nju.edu.cn

water dissociation, thereby facilitating the efficient supply of H[*10,11]. Using this strategy, numerous pH-universal HER electrocatalysts have been reported (Supplementary Tables 1, 2). Although a few of them have shown HER overpotentials less than 30 mV at a benchmark current density of 10 mA cm[-2], the pH-universal catalysts with low overpotentials (< 250 mV) at industrial-current density (≥ 1 A cm[-2]) in all-pH electrolytes have not yet been realized to date[12], due to the difficult creation of a densely atomically mixed interface to form H[*].

As we know, for proton-blocking metal catalysts like Ru, Pt and Ir, the proton reduction and HER occur on the surface. The shift in HER mechanism with local pH restricts the development of pH-universal catalysts[13,14]. In contrast, for non-proton-blocking materials such as certain transition oxides ($WO_3$, $MoO_3$, etc.), proton can be electrochemically inserted into the crystal lattice in all-pH electrolytes, forming a "hydrogen reservoir"[15–17]. It is anticipated that if rapid proton migration pathways can be established from non-proton-blocking support to proton-blocking metal catalysts, the impact of the pH value on the H[*] formation can be mitigated or even eliminated, thereby achieving a competitive pH-universal HER performance. With this consideration, we construct a HER catalyst by loading the proton-blocking Ru nanoparticles (NPs) onto the non-proton-blocking $H_xWO_3$ nanoneedles (NN) obtained by thermal hydrogenation of $WO_3$ NN, denoted as Ru-$H_xWO_3$ NN. Combining the in-situ experimental results

with theoretical simulations, we demonstrate a lattice-hydrogen cycling mechanism: the $H_xWO_3$ NN with abundant lattice hydrogen functions as a "hydrogen reservoir" that can efficiently supply hydrogen species to the Ru sites via a rapid lattice-hydrogen migration. Meanwhile, the consumed lattice-hydrogen in $H_xWO_3$ NN is spontaneously replenished in all-pH electrolytes through hydrogen adsorption (acidic) or water dissociation (alkaline/neutral). Such a unique mechanism confers the Ru-$H_xWO_3$ NN with a competitive pH-universal HER performance, achieving low overpotentials of 125, 219, and 142 mV at an industrial-level current density of 1 A cm[-2], along with great durability (500 h@1 A cm[-2]) in 0.5 M $H_2SO_4$, 1 M phosphate buffered solution (PBS), and 1 M KOH, respectively.

## Result

### Structural characterization of Ru-$H_xWO_3$ NN and Ru-$WO_3$ NN

The preparation and characterizations of Ru-$H_xWO_3$ NN via thermal-hydrogenation are shown in Fig. 1. For comparison, the synthesis of Ru-$WO_3$ NN via $NaBH_4$-reduction is also presented (Supplementary Fig. 1, and see details in Methods). Scanning electron microscopy (SEM) and transmission electron microscopy (TEM) observations reveal that $WO_3$ NN are densely grown on the surface of copper foam (CF), with typical tip diameter of ~25 nm. Specifically, the $WO_3$ NN shows the lattice distances of 0.385 and 0.639 nm for the (002) and (100) planes,

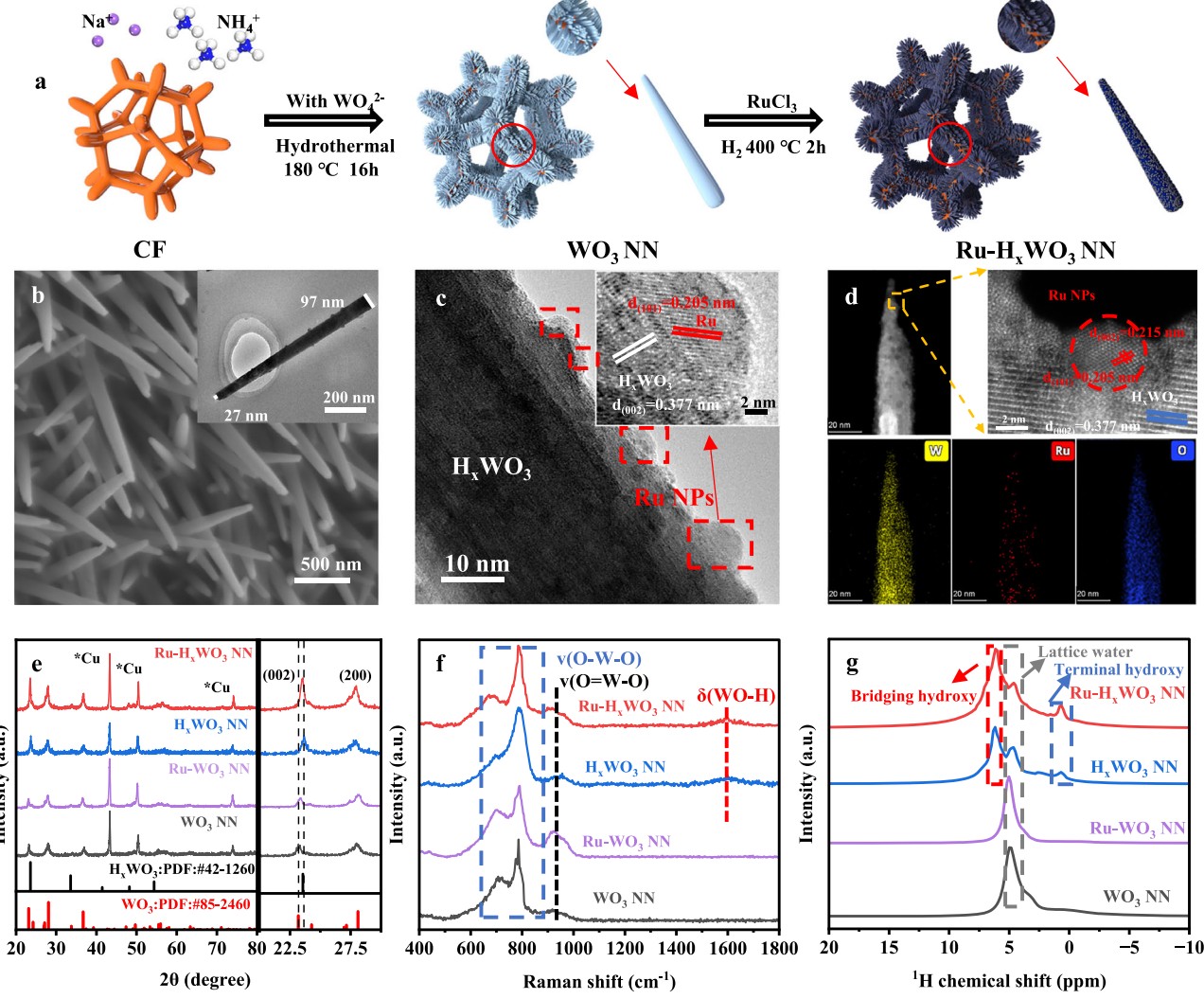

**Fig. 1 | Preparation and characterizations of Ru-$H_xWO_3$ NN. a** Schematic preparation route. **b,c** SEM (**b**) and HRTEM images (**c**). Insets are the corresponding TEM image and local enlargement. **d** HAADF-STEM image and corresponding EDS elemental mappings of W, Ru, and O. **e** XRD pattern and local enlargement in the range of 20°−30°. **f** Raman spectrum. **g** ¹H NMR spectrum. Note: The spectra for $WO_3$ NN, $H_xWO_3$ NN and Ru-$WO_3$ NN are also shown in (**e−g**) for comparison.

respectively. After undergoing $H_2$ thermal treatment, the resulting $H_xWO_3$ NN shows a smaller interplanar distance of 0.377 nm for the (002) planes (Supplementary Fig. 2). This phenomenon indicates that hydrogen is inserted into the interlayer galleries of the (002) planes during the thermal reduction process, which in turn leads to an obvious shrinkage of the $WO_3$ lattice. CV tests and thermogravimetric analysis (TGA) further confirm the insertion of hydrogen into $WO_3$ NN, with x value of ~0.8 in $H_xWO_3$ NN (Supplementary Figs. 3 and 4). It is worth noting that when protons are inserted into $WO_3$ to form $H_yWO_3$ by electrochemical method, the typical value of y is ca. 0.5, which is significantly lower than that in the thermally-hydrogenated $H_xWO_3$ NN (Supplementary Fig. 3)[13].

For Ru-$H_xWO_3$, the nanoneedle morphology is well-preserved, and hemispherical Ru NPs are highly dispersed on the surface of $H_xWO_3$ (Fig. 1b,c and Supplementary Fig. 5). The (101) planes of Ru NPs (with a lattice distance of 0.205 nm) and the (002) facets of $H_xWO_3$ (with a lattice distance of 0.377 nm) form an intersection angle of 45°, showing a good lattice match at the heterogenous interfaces. The high-angle annular dark field scanning transmission microscopy (HAADF-STEM) image of a Ru-$H_xWO_3$ NN, along with the corresponding energy dispersive X-ray spectroscopy (EDS) elemental mapping images, verifies the homogeneous distribution of tungsten (W) and oxygen (O), as well as the dotlike distribution of Ru, which is consistent with the observations from high-resolution transmission electron microscopy (HRTEM) (Fig. 1d). In the case of Ru-$WO_3$, the morphology of the small Ru NPs immobilized on the surface of $WO_3$ NN is similar to that of Ru-$H_xWO_3$ NN, but the distance between the (002) planes of $WO_3$ NN remains at 0.385 nm. This result confirms that the reduction by $NaBH_4$ only led to the formation of metallic Ru NPs but did not result in the insertion of hydrogen into $WO_3$ (Supplementary Fig. 6).

X-ray diffraction (XRD) analysis shows that, after thermal-hydrogenation, the (002) peak shifts positively from 23.2° to 23.6°. The in situ XRD patterns show that the (002) peak gradually shifts to high angles during the temperature-increasing process, and the position of (002) peak keeps constant in the subsequent cooling process, which confirms the formation of $H_xWO_3$ during the thermal-hydrogenation (Supplementary Fig. 7). This shift also reflects the shrinkage of the (002) lattice after hydrogen insertion, which is in line with the HRTEM observations (Fig. 1e, and Supplementary Figs. 2 and 5). When comparing the Raman spectra of $H_xWO_3$ and Ru-$H_xWO_3$ NN with those of $WO_3$ and Ru-$WO_3$ NN, a new signal appears at approximately 1580 $cm^{-1}$, which corresponds to the bending vibration band of WO-H bond [δ(WO-H)] (Fig. 1f)[18,19]. The solid-state $^1H$ nuclear magnetic resonance ($^1H$ NMR) spectra of $H_xWO_3$ and Ru-$H_xWO_3$ NN indicate the emergence of bridging hydroxy (H-$O_{Bridging}$, in bulk) and terminal hydroxy (H-$O_{Terminal}$, on surface) groups, accompanied by the weakened signal of lattice water (Fig. 1g)[20]. These results clearly confirm the formation of lattice-hydrogen in $H_xWO_3$ and Ru-$H_xWO_3$ NN through the $H_2$ thermal treatment.

The electronic structure characteristics of Ru-$H_xWO_3$ NN, Ru-$WO_3$ NN, $WO_3$ NN, and $H_xWO_3$ NN are presented in Fig. 2. X-ray photoelectron spectroscopy (XPS) analysis reveals the existence of $W^{6+}$ and $Ru^0$ species in Ru-$H_xWO_3$ NN and Ru-$WO_3$ NN, with the close Ru contents of 1.21 and 1.26 wt.%, respectively (Fig. 2a, b and Supplementary Table 3). The binding energy (BE) of the $W^{6+}$ species in $WO_3$ NN is 36.15 eV. In contrast, the BE of the $W^{6+}$ species negatively shifts to 35.80 eV in $H_xWO_3$ NN due to hydrogen insertion, and further negatively shifts to 35.65 eV in Ru-$H_xWO_3$ NN due to the additional loading of Ru. When only Ru is loaded (i.e., without hydrogen insertion), the corresponding BE slightly negatively shifted to 36.05 eV in Ru-$WO_3$ NN[21] (Fig. 2a). Regarding the Ru 3p spectra, when compared with commercial Ru/C, the $Ru^0$ signal in Ru-$H_xWO_3$ and Ru-$WO_3$ NN shows positive BE shifts of 0.20 and 0.15 eV, respectively[22] (Fig. 2b). The XPS results reflect an electron transfer from Ru to $H_xWO_3$ NN and $WO_3$ NN.

X-ray absorption fine structure (XAFS) spectra were employed to further reveal chemical state of W and Ru (Fig. 2c, d). The W $L_3$-edge X-ray absorption near-edge structure (XANES) spectra indicate that the edge energy changes in the order of Ru-$H_xWO_3$ NN <$H_xWO_3$ NN <Ru-$WO_3$ NN < $WO_3$ NN. The Ru K-edge XANES spectra show that the energy absorption threshold values of Ru-$H_xWO_3$ NN and Ru-$WO_3$ NN are slightly higher than that of Ru foil with the former being more positive. These results are in agreement with XPS results (Fig. 2a–d). The XAFS results also suggest a strong interaction between Ru and $WO_3$ or $H_xWO_3$ NN, with electron transfer from $Ru^0$ to $W^{6+}$. The W $k^2$-weighted R-space Fourier transform extended X-ray absorption fine structure (FT-EXAFS) spectra for all samples display a main peak around 1.40 Å for the W-O bond, without the appearance of the W-W signal around 2.75 Å (Fig. 2e). The Ru $k^2$-weighted R-space FT-EXAFS spectra for Ru-$H_xWO_3$ NN and Ru-$WO_3$ NN show two peaks around 1.55 and 2.40 Å, which correspond to Ru-O and Ru-Ru bonds, respectively. These results and corresponding theoretical calculations confirm that the Ru NPs bond to the $H_xWO_3$ and $WO_3$ NN through Ru-O ionic bonds, rather than Ru-W metallic bonds (Fig. 2f, and Supplementary Fig. 8).

Accordingly, a model of Ru-$H_xWO_3$ was constructed to calculate the work function and charge difference density (Fig. 2g). Density functional theory (DFT) calculations indicate that the work functions of the Ru (101) plane and the $H_xWO_3$ (002) plane are 4.75 and 5.17 eV, respectively. The lower work function of Ru than $H_xWO_3$ suggests a tendency of electron transfer from Ru to $H_xWO_3$. The charge difference density analysis corroborates this result, showing electron accumulation at the $H_xWO_3$ surface and electron depletion at the Ru plane (Fig. 2g). This electron redistribution causes the d-band centers of Ru and W in the Ru-$H_xWO_3$ system to shift closer to the Fermi level ($E_f$) compared to those in Ru-$WO_3$, which can enhance the adsorption stability of key intermediates (e.g., $H^*$, $H_2O^*$) on Ru-$H_xWO_3$[23]. Moreover, the higher Fermi level occupancy ($E-E_f = 0$) for Ru and W in Ru-$H_xWO_3$ further facilitates the electron conductivity and HER thereof (Fig. 2h, i)[24]. Overall, compared with Ru-$WO_3$ NN, the thermally-hydrogenated $H_xWO_3$ NN induces more electron transfer from Ru to $H_xWO_3$ NN, leading to enhanced electron conductivity and improved intermediate adsorption in Ru-$H_xWO_3$ NN.

## Electrocatalytic HER performances

The HER performances of Ru-$H_xWO_3$ NN, Ru-$WO_3$ NN, $H_xWO_3$ NN, along with the benchmark catalysts Pt/C and Ru/C, were systematically examined, as illustrated in Fig. 3. In 0.5 M $H_2SO_4$, 1 M PBS, and 1 M KOH electrolytes, Ru-$H_xWO_3$ NN exhibits low overpotentials of 12 mV, 28 mV, and 14 mV, respectively, at a current density of 10 mA $cm^{-2}$. These values are significantly lower than those of Pt/C and Ru/C (Fig. 3a–c). In contrast, under the same conditions, Ru-$WO_3$ NN displays notably higher overpotentials of 45, 80, and 53 mV, while $H_xWO_3$ NN shows the high overpotentials of 140, 148, and 151 mV, respectively. This result clearly demonstrates the lattice hydrogen in $H_xWO_3$ NN plays a crucial role in the great HER performance of Ru-$H_xWO_3$ NN due to the heterogenous coupling between Ru and $H_xWO_3$ NN (Fig. 3a-c and Supplementary Fig. 9). When normalized by the electrochemical active surface area (ECSA), Ru-$H_xWO_3$ NN also shows the best pH-universal performance among the examined catalysts, indicating its optimal intrinsic activity for the HER (Supplementary Figs. 10 and 11). Notably, Ru-HxWO3 NN exhibits mass activities at 100 mV that are competitive with common benchmarks (Pt/C and Ru/C), consistent with efficient utilization of noble metals (Supplementary Fig. 12). Electrochemical impedance spectroscopy (EIS) analysis reveals that Ru-$H_xWO_3$ NN has the smallest charge-transfer resistance ($R_{ct}$) among all the catalysts in all-pH electrolytes, reflecting its accelerated HER kinetics (Supplementary Fig. 13).

The Ru-$H_xWO_3$ NN catalyst demonstrates a competitive pH-universal HER performance even at an ampere-level current density of

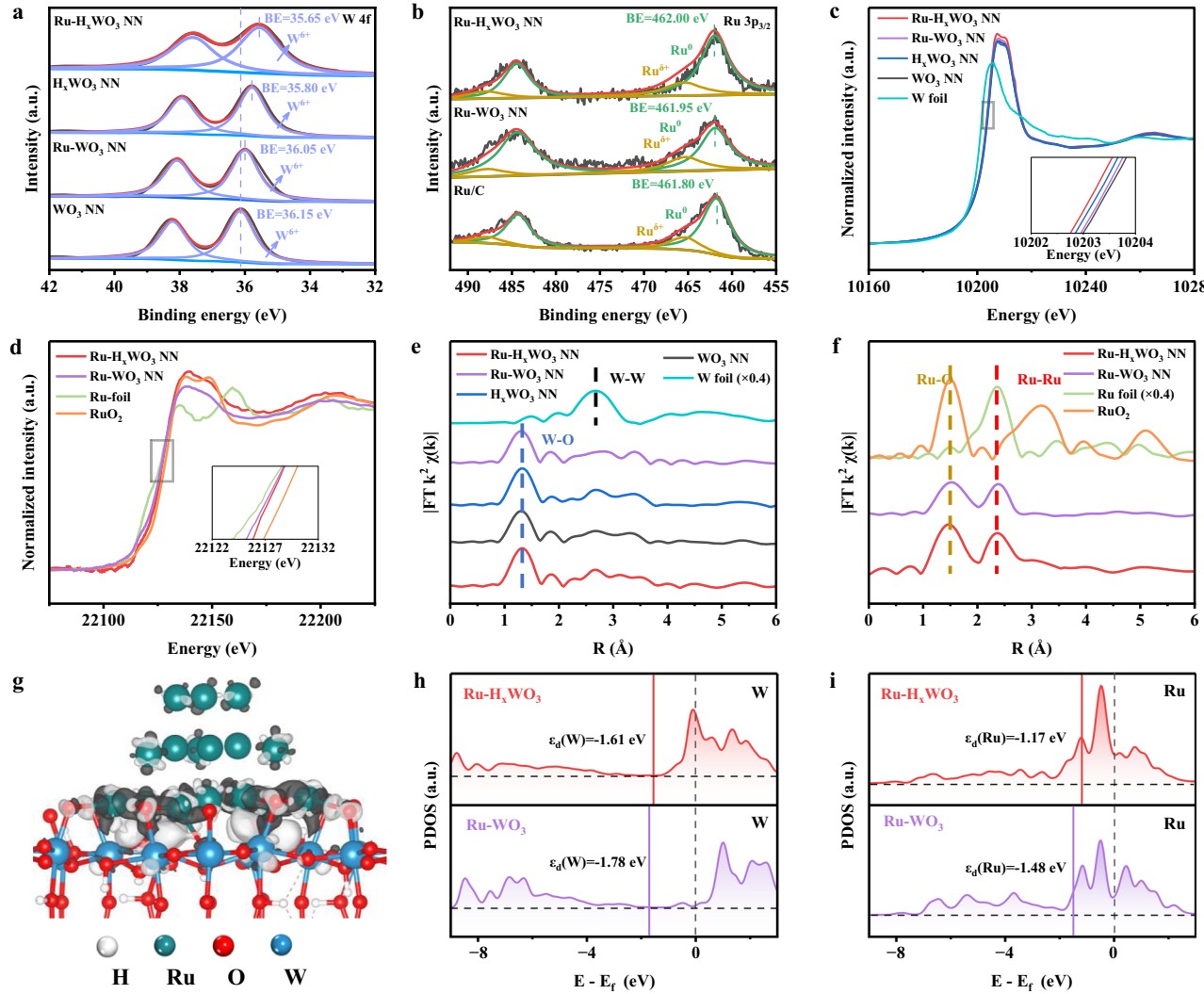

**Fig. 2 | Electronic structure characteristics of Ru-H$_x$WO$_3$ NN, Ru-WO$_3$ NN, WO$_3$ NN, and H$_x$WO$_3$ NN. a,b** W 4$f$ and Ru 3$p_{3/2}$ spectra, respectively. For comparison, the spectrum for commercial Ru/C is presented in (**b**). **c,d** W L$_3$-edge and Ru K-edge XANES spectra, respectively. For comparison, the spectrum for W foil is presented in (**c**), and those for Ru foil and RuO$_2$ are presented in (**d**). The marks in (**c**) and (**d**) is local enlargement. **e,f** Corresponding $k^2$-weighted R-space Fourier transformed EXAFS spectra for W (**e**) and Ru (**f**). **g** Charge difference density. Black and white regions represent electron depletion and accumulation, respectively. The isosurface value is set at 0.01. **h,i** The PDOS curves of 3 $d$ orbitals of W (**h**) and Ru (**i**) atoms in Ru-H$_x$WO$_3$ NN and Ru-WO$_3$ NN. The $d$-band centers of the corresponding metal are marked in black font.

1 A cm$^{-2}$, with the low overpotentials of 125 mV, 219 mV, and 142 mV in 0.5 M H$_2$SO$_4$, 1 M PBS and 1 M KOH electrolytes, respectively (Supplementary Fig. 9). Attractively, an extraordinary long-term durability for over 500 h is achieved at 1 A cm$^{-2}$ in all-pH electrolytes, with negligible changes in overpotentials and polarization curves before and after the chronopotentiometry (CP) tests (Fig. 3d and Supplementary Fig. 14). Furthermore, after the durability tests in all-pH electrolytes, the morphology, composition and structure of Ru-H$_x$WO$_3$ NN remain almost identical to those of the pristine catalyst, indicating its high structural stability and electrocatalytic stability (Supplementary Figs. 15, 16 and Table 4).

Moreover, among the examined catalysts, the Ru-H$_x$WO$_3$ NN exhibits the lowest Tafel slopes, with values of 28.8, 34.5, and 34.2 mV dec$^{-1}$ in 0.5 M H$_2$SO$_4$, 1 M PBS and 1 M KOH, respectively. In contrast, the Tafel slopes of Ru-WO$_3$ NN in the same electrolytes are 55.8, 86.5, and 82.6 mV dec$^{-1}$, respectively, which are close to those of Ru/C (Supplementary Fig. 17). This result implies that the rate-determining step (RDS) for Ru-H$_x$WO$_3$ NN is the Tafel process, while those for Ru-WO$_3$ NN and Ru/C are the Heyrovsky process[25]. This disparity can be

attributed to the significant enhancement effect of the H$_x$WO$_3$ NN on the HER performance of Ru NPs.

Generally speaking, Ru-H$_x$WO$_3$ NN exhibits the most competitive pH-universal HER performance among the examined catalysts (Fig. 3d, and Supplementary Table 1 and Fig. 18). Notably, it is also competitive with most HER catalysts in single electrolytes at ampere-level current density (Supplementary Table 2).

## The HER enhancement mechanism of Ru-H$_x$WO$_3$ NN
**The HER characteristics of the H$_x$WO$_3$ support.** We first investigated the characteristics of the HER of the H$_x$WO$_3$ support itself (x = 0.88 ~ 0.97). The rotating ring-disk electrode (RRDE) technique was employed to quantitatively monitor the local pH on the H$_x$WO$_3$ NN surface at different potentials in 0.1 M PBS solution (pH=7.02) (Supplementary Figs. 19 and 20)[26]. The results were compared with those of WO$_3$ NN and glassy carbon (GC). It was found that the pH on GC remained nearly constant at ~7.0 within the potential range of 0.1 ~ -0.5 V. In contrast, the local pH on the surface of H$_x$WO$_3$ NN was approximately 3.7 within 0.1 ~ -0.2 V, indicating the presence of a local

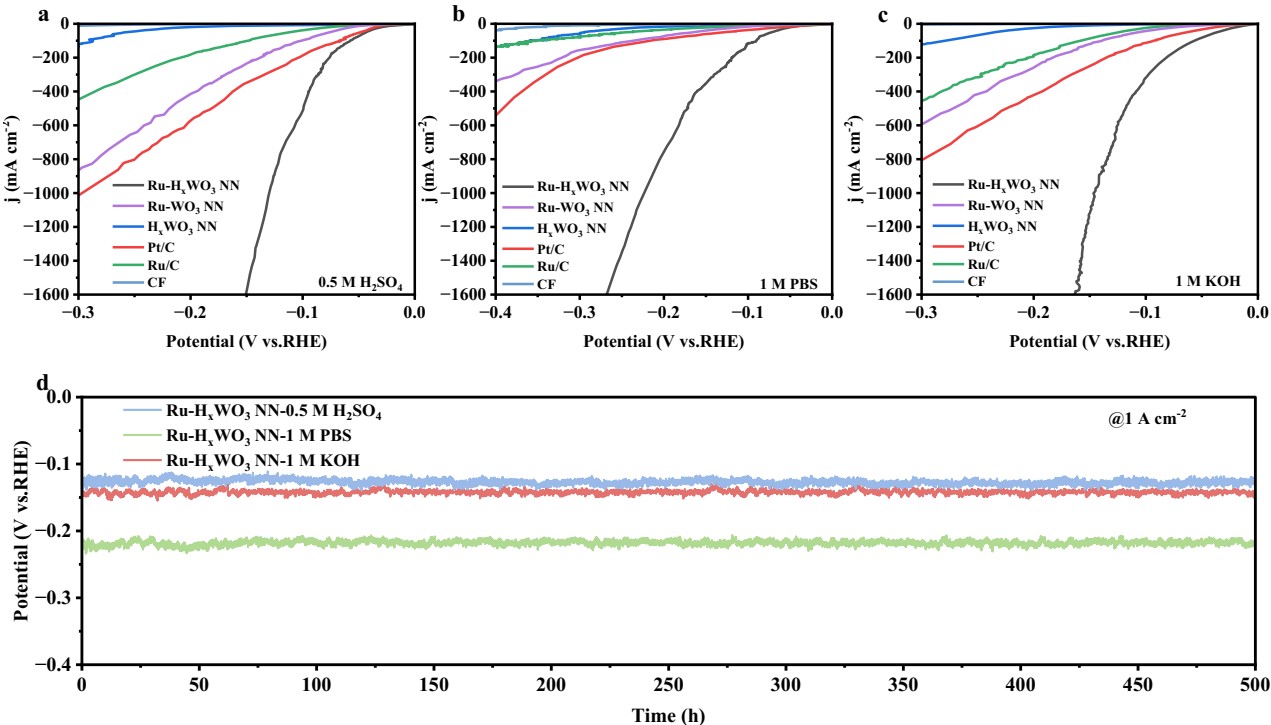

**Fig. 3 | pH-universal HER performances of Ru-H$_x$WO$_3$ NN, Ru-WO$_3$ NN, and H$_x$WO$_3$ NN in 0.5 M H$_2$SO$_4$, 1 M PBS and 1 M KOH. a–c** Polarization curves. **d** CP curves of Ru-H$_x$WO$_3$ NN at the current density of 1 A cm$^{-2}$ for 500 h. Note: The data of Pt/C, Ru/C and CF in (**a–c**) are presented for comparison.

acid-like microenvironment due to the thermally-inserted hydrogen in H$_x$WO$_3$ NN (Fig. 1f,g)[17,27]. The local pH on the surface of WO$_3$ NN decreased from 7.58 (weak basicity) to 5.72 (weak acidity) as the potential shifted negatively from 0.1 V to -0.3 V, due to the electrochemical insertion of hydrogen into WO$_3$ to form H$_y$WO$_3$ (y = -0.5) (Fig. 4a)[13]. The lower local pH on H$_x$WO$_3$ NN than on WO$_3$ NN indicates that thermal H insertion can result in a higher degree of hydrogenation of WO$_3$ than electrochemical H insertion. This situation leads to a much higher HER performance of Ru-H$_x$WO$_3$ NN than Ru-WO$_3$ NN (Supplementary Fig. 21). As the potential is further shifted negatively, the local pH values gradually increase due to the consumption of H species during the HER. For both H$_x$WO$_3$ NN and WO$_3$ NN, the lattice-H can migrate between the adjacent oxygen sites. The energy barriers for this migration are much lower in H$_x$WO$_3$ than in WO$_3$, indicating the enhanced lattice-H migration kinetics in H$_x$WO$_3$ (Fig. 4b and Supplementary Fig. 22). After removing the bias, the pH on H$_x$WO$_3$ NN can return to a low value (Supplementary Fig. 20e). This result reflects that the lattice-H species in H$_x$WO$_3$ can be spontaneously replenished to go back to the local acidity. Similar phenomena were also observed in H$_2$SO$_4$ and KOH electrolytes, demonstrating the formation of local acid-like microenvironment on the H$_x$WO$_3$ surface under all-pH conditions (Supplementary Figs. S19 and S20).

**The enhanced HER activity of Ru-H$_x$WO$_3$ NN.** To verify the participation of lattice-H in the HER of H$_x$WO$_3$ NN, in situ Raman spectroscopy was used to monitor the HER process of Ru-H$_x$WO$_3$ NN in 1 M PBS isotope deuteroxide (D$_2$O) solution at -0.1 V (Fig. 4c). At the open circuit potential (OCP), the δ(WO-H) peak at ~1580 cm$^{-1}$ was observed as expected. On applying the potential of -0.1 V (10 s), the stretching vibration peak of ν(Ru-H) at ~872 cm$^{-1}$ appeared[28], accompanied by the weakening of the δ(WO-H) peak. After 60 s' testing, only trace of the ν(Ru-H) and δ(WO-H) peaks remained, and two new peaks of ν(Ru-D) at ~600 cm$^{-1}$ and δ(WO-D) at ~1200 cm$^{-1}$ were observed[19]. At 300 s, the ν(Ru-H) and δ(WO-H) peaks completely disappeared, leaving the ν(Ru-

D) and δ(WO-D) signals. It is seen, during this process, the δ(WO-H)/ν(Ru-H) is gradually replaced by the δ(WO-D)/ν(Ru-D). When returning OCP condition, only δ(WO-D) signal could be detected. Similar process was also observed in 1 M KOH electrolyte with D$_2$O solvent (Supplementary Fig. 23). These results confirm the dynamic migration of lattice-H from the H$_x$WO$_3$ support to Ru for the HER, and the replenishment of lattice-D through the electrochemical insertion of deuterium. In addition, the kinetic isotope effect (KIE) was examined in either 1 M PBS or 1 M KOH electrolyte with H$_2$O or D$_2$O solvent[29]. The KIE values ($J_{H2O}/J_{D2O}$) remained constant for Ru-WO$_3$ NN, while increased rapidly for Ru-H$_x$WO$_3$ NN, reflecting the migration and reconstruction of lattice-H/D during the HER for Ru-H$_x$WO$_3$ NN (Supplementary Fig. 24). Therefore, the isotope experiment results clearly prove the involvement of pre-inserted lattice-H and in situ electrochemical replenishment of lattice-H/D during the HER of Ru-H$_x$WO$_3$ NN.

To gain a deeper understanding of the enhanced HER activity of Ru-H$_x$WO$_3$ NN catalyst, in situ Raman spectra were recorded. The potential was increased negatively from 0 to -0.3 V and then returned to 0 V, with a potential interval of 0.05 V. In 1 M PBS solution, for the H$_x$WO$_3$ NN support, the ν(W-O) peaks at 698, 790, and 920 cm$^{-1}$ became weaker/stronger as the potential was increased negatively and then recovered, respectively, while the δ(WO-H) peak at 1580 cm$^{-1}$ showed an opposite trend of change[18,30] (Fig. 4d). This result indicates that the lattice-H concentration in H$_x$WO$_3$ can be further regulated by the electrochemical process[17]. In contrast, for the Ru-H$_x$WO$_3$ NN, both the ν(W-O) and δ(WO-H) peaks of H$_x$WO$_3$ became weaker/stronger as the potential was increased negatively and then recovered, respectively. Meanwhile, a new ν(Ru-H) peak emerged, and became stronger/weaker as the potential was increased negatively and then recovered, correspondingly (Fig. 4e). These results clearly indicate the lattice-H in H$_x$WO$_3$ NN [δ(WO-H)] can dynamically migrate to the Ru NPs [ν(Ru-H)] under the applied potential, meanwhile the lattice-H can be in situ replenished electrochemically. The weakening of ν(W-O) peaks is directly associated with the migration of lattice-H via hopping on

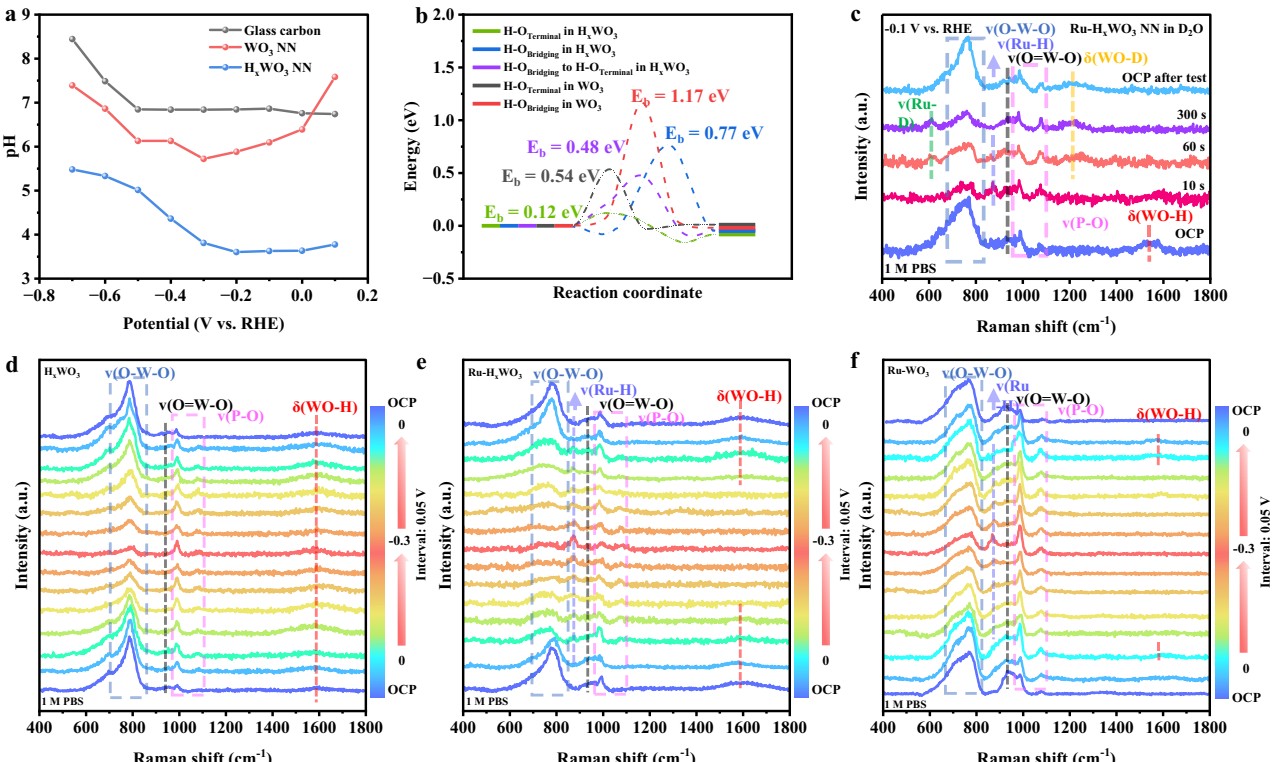

**Fig. 4 | The behavior of lattice-hydrogen in Ru-H$_x$WO$_3$ NN and H$_x$WO$_3$ NN.**
**a** Local pH values on the surfaces of H$_x$WO$_3$ NN, WO$_3$ NN and GC at different potentials in 0.1 M PBS with pH of 7.02. **b** Calculated energy barrier diagram for H migration. **c** In situ Raman spectra of Ru-H$_x$WO$_3$ NN at -0.1 V in 1 M PBS D$_2$O solution at different stages of HER. **d–f** In situ Raman spectra of H$_x$WO$_3$ (**d**), Ru-H$_x$WO$_3$ (**e**) and Ru-WO$_3$ (**f**) NN in 1 M PBS solution at different potentials. Note: The Raman signals at ~985 and ~1075 cm$^{-1}$ are attributed to the symmetric P-O stretching mode of PO$_4^{3-}$ from the electrolyte[42].

lattice oxygen (Fig. 4b). As a comparison, for Ru-WO$_3$ without pre-inserted lattice-H, a trace of the δ(WO–H) and ν(Ru-H) peaks could be observed at -0.05 V, which is associated to the electrochemical insertion of hydrogen into WO$_3$ and its subsequent migration to the Ru NPs. Then, the Raman spectra of Ru-WO$_3$ show a similar evolution to those of Ru-H$_x$WO$_3$ NN (Fig. 4f). The residual intensity of ν(W-O) peaks at -0.3 V is much stronger in Ru-WO$_3$ than in Ru-H$_x$WO$_3$. These results indicate that, compared to Ru-H$_x$WO$_3$, Ru-WO$_3$ has a lower lattice-H concentration from electrochemical H insertion and less dynamic migration of lattice-H to Ru. This situation leads to a higher intensity of the ν(Ru-H) signal in Ru-H$_x$WO$_3$ than in Ru-WO$_3$, i.e., higher H$^*$ coverage on Ru for the former (Supplementary Fig. 25). Similar results were also obtained in H$_2$SO$_4$ and KOH electrolytes (Supplementary Figs. 25 and 26). The higher H$^*$ coverage on Ru in Ru-H$_x$WO$_3$ than in Ru-WO$_3$ in all-pH electrolytes is responsible for its enhanced pH-universal HER performance. Our in situ EIS study on the H adsorption behaviors at different overpotentials also support this conclusion (Supplementary Figs. 27–29 and Tables 5–7)[31,32].

**The cycling of hydrogen migration, evolution, and replenishment**

To further understand the hydrogen migration, evolution, and replenishment during the HER process, DFT calculations were conducted based on the optimized models of WO$_3$, H$_x$WO$_3$, Ru-WO$_3$ and Ru-H$_x$WO$_3$, as shown in Fig. 5 (Supplementary Fig. 30). First, the free energies of H adsorption (ΔG$_{H^*}$) on all possible active sites are calculated (Fig. 5a–c). For WO$_3$, both the terminal and bridging O sites on its surface display highly negative H adsorption free energies (Supplementary Fig. 31), which results in the easy H insertion but poor HER

performance. In the case of Ru-WO$_3$, the ΔG$_{H^*}$ values of all Ru sites are positive (Fig. 5b, c). The ΔG$_{H^*}$ for the terminal O atoms connecting with Ru (i.e., Ru-O) and the bridging O atoms near Ru (O$_{Bridging-I}$) turn positive, while those for both the terminal O and bridging O sites away from Ru (i.e., O$_{Terminal}$ and O$_{Bridging-II}$) remain negative (Fig. 5b). Consequently, the H atom preferentially adsorbs on the WO$_3$ support rather than on Ru sites. This result aligns with the inferior lattice-H migration ability of the WO$_3$ support and the relatively poor HER performance of Ru-WO$_3$ (Figs. 4b and 3d). On the contrary, for Ru-H$_x$WO$_3$, which contains abundant lattice-H atoms, all O sites exhibit positive ΔG$_{H^*}$, which is conducive to the migration of lattice-H atoms to the Ru sites (Fig. 5b). Moreover, the ΔG$_{H^*}$ for the Ru$_{Interfacial}$ in H$_x$WO$_3$ is -0.08 eV, very close to the ideal value for HER. In contrast, the ΔG$_{H^*}$ for the possible Ru and W active sites remain highly positive, showing the inferior HER activity of these sites (Fig. 5b,c, Supplementary Fig. 32)[33]. As a result, a feasible pathway for the lattice-H migration is established at the interface of Ru-H$_x$WO$_3$, which involves the lattice-H migration from the O$_{Bridging-I}$ to the terminal O, and further to the interfacial Ru sites (i.e., step 1 and step 2 in Fig. 5a). Through Climbing Image Nudged Elastic Band (CI-NEB) calculations, it was determined that the kinetic migration energy barriers in Ru-H$_x$WO$_3$ are quite small, only 0.21 eV in step 1 and 0.20 eV in step 2, which are significantly lower than the corresponding 0.78 and 1.25 eV in Ru-WO$_3$, respectively (Fig. 5d). This finding indicates that, in Ru-H$_x$WO$_3$, it is easy to realize the lattice-H migration to the highly active HER sites, i.e., the interfacial Ru sites with a near-zero ΔG$_{H^*}$ (-0.08 eV), which leads to competitive HER performance (Fig. 3)[33].

The lattice-H atoms consumed at the interfacial Ru can be readily replenished by the rapid hydrogen migration from the WO-H sites

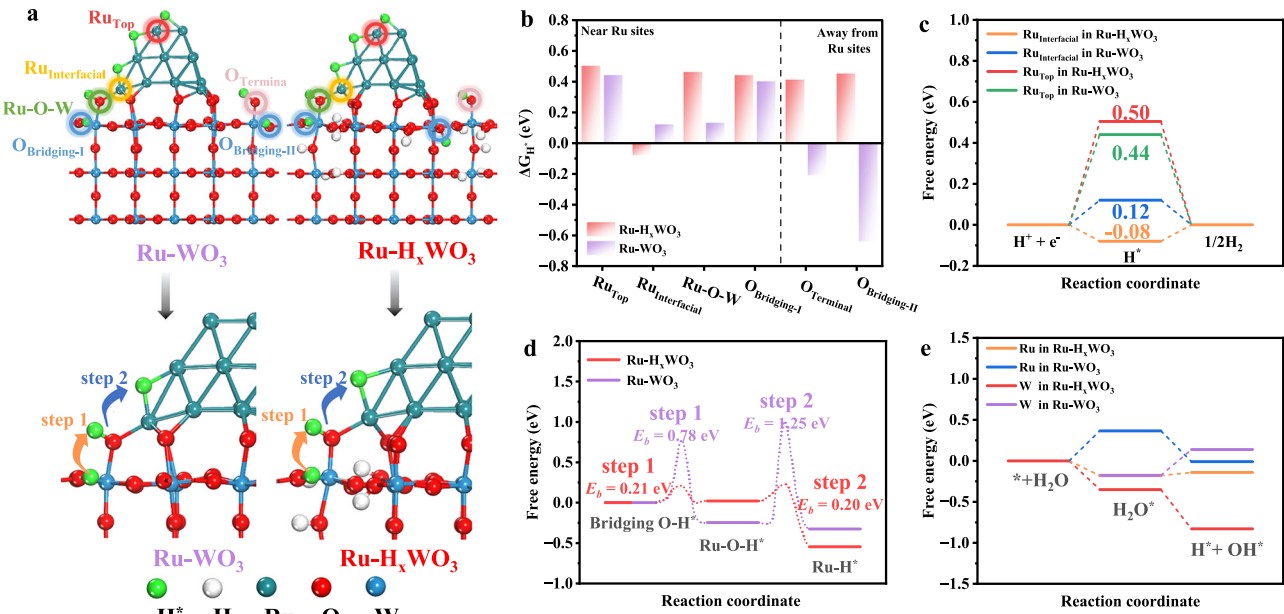

**Fig. 5 | DFT calculation of the dynamic migration and replenishment of lattice-H. a,b** Schematic diagram of different sites for H adsorption and migration near the interface (**a**) and corresponding adsorption free energy (**b**) in Ru-WO$_3$ and Ru-H$_x$WO$_3$. **c** Free energy profiles for HER on Ru sites in Ru-H$_x$WO$_3$. **d** Hydrogen migration energy barriers corresponding to step 1 and step 2 in (**a**). **e** Free energy profiles for water dissociation on Ru and W sites in Ru-WO$_3$ and Ru-H$_x$WO$_3$.

away from the Ru sites via H hopping between O sites with low energy barriers (e.g., <0.6 eV) (Fig. 4b, and Supplementary Fig. 22)[34]. The lattice-H atoms can be sourced from the electrolytes. In acidic electrolyte, due to the strong affinity between H$^+$ ions and the O sites of H$_x$WO$_3$, H$^+$ ions can adsorb onto these sites to form lattice-H atoms (Supplementary Fig. 31). In alkaline or neutral electrolytes, the lattice-H is replenished through water dissociation, which is, however, often hindered by sluggish kinetics. To understand this phenomenon, we calculated the water dissociation process on each metal site in Ru-WO$_3$ and Ru-H$_x$WO$_3$ (Supplementary Fig. 31). The results show that for Ru-WO$_3$, H$_2$O molecules tend to adsorb on the W sites rather than the Ru sites. In contrast, for Ru-H$_x$WO$_3$, H$_2$O molecules can spontaneously adsorb on both Ru and W sites, with the W sites having a more negative adsorption energy, which is confirmed by the contact angle (Supplementary Fig. 33). Unlike the W sites in Ru-WO$_3$ and the Ru sites in Ru-H$_x$WO$_3$, the W sites in Ru-H$_x$WO$_3$ can spontaneously dissociate the adsorbed water, as evidenced by its downhill trend of the free energy profile (Fig. 5e). Thus, the lattice-H in Ru-H$_x$WO$_3$ can be easily replenished.

The preceding experimental results, supported by theoretical studies, validate a lattice-H cycling mechanism for the HER of Ru-H$_x$WO$_3$ NN. The H$_x$WO$_3$ NN support with abundant lattice-H obtained by thermal hydrogenation of WO$_3$ NN can serve as a "H reservoir" and continuously supply H species to the highly-active HER sites of the interfacial Ru sites ($\Delta G_{H^*}$ = -0.08 eV), leading to the competitive HER performance via the Tafel process (Supplementary Fig. 17). The consumed lattice-H in H$_x$WO$_3$ support is spontaneously replenished in all-pH electrolytes through H adsorption (acidic) or water dissociation (alkaline/neutral). A rapid lattice-H migration pathway is established from non-proton-blocking support (H$_x$WO$_3$) to proton-blocking metal catalysts (Ru) through H hopping between O sites. Such a process reduces dependence on the pH of the used electrolytes. This strategy should also be applicable to some other combination of non-proton-blocking support with proton-blocking metal catalysts. In fact, by replacing Ru with Ir or Pt, the corresponding Ir-H$_x$WO$_3$ NN and Pt-H$_x$WO$_3$ NN did exhibit much better pH-universal HER performances than Ir-WO$_3$ NN and Pt-WO$_3$ NN, respectively (Supplementary Fig. 34).

## Discussion
This study addresses the critical challenge of designing pH-universal HER electrocatalysts capable of operating efficiently at industrial current densities. By a lattice-hydrogen cycling mechanism, we decouple H$^*$ availability from electrolyte pH, enabling robust HER performance across acidic, neutral, and alkaline environments. The integration of proton-blocking Ru nanoparticles with thermally-hydrogenated H$_x$WO$_3$ nanoneedles creates a dynamic hydrogen reservoir, where pre-inserted lattice hydrogen in H$_x$WO$_3$ migrates swiftly to Ru active sites via low-energy pathways (0.20–0.21 eV). Meanwhile, the consumed hydrogen is replenished through proton adsorption (acidic) or water dissociation (alkaline/neutral), ensuring sustained catalytic activity. In situ Raman spectroscopy, isotopic labeling, and DFT calculations confirm the lattice-hydrogen migration and replenishment processes. As a result, the Ru-H$_x$WO$_3$ catalyst achieves a competitive pH-universal HER performance with low overpotentials of 125 mV (0.5 M H$_2$SO$_4$), 219 mV (1 M PBS), and 142 mV (1 M KOH) at 1 A cm$^{-2}$, alongside 500-hour stability. The broader utility of the design strategy for pH-universal catalysts is also demonstrated by the applicability to Ir and Pt-based systems. Further attention should be paid to the validation of the mechanism's efficacy in other hydrogen-involving reactions, such as CO$_2$RR, NRR, and other renewable energy-driven processes. Additionally, real-world conditions such as fluctuating renewable energy inputs and seawater electrolysis are expected to be explored.

In a short, this work establishes a groundbreaking strategy for pH-robust electrocatalysts, bridging fundamental insights with industrial relevance. By overcoming pH-dependent kinetic limitations, the lattice-hydrogen cycling mechanism paves the way for sustainable energy conversion systems, aligning with global decarbonization goals.

## Methods
### Materials
Sodium tungstate dihydrate (Na$_2$WO$_4$·2H$_2$O, AR), ammonium sulfate ((NH$_4$)$_2$SO$_4$, AR), sodium sulfate (Na$_2$SO$_4$, AR), and deuteroxide (D$_2$O, AR) were purchased from Shanghai Macklin Biochemical Co., Ltd. Potassium phosphate monobasic (KH$_2$PO$_4$, AR) and potassium

hydrogen phosphate ($K_2HPO_4$, AR) were obtained from Shanghai Aladdin Bio-Chem Technology Co., Ltd. Potassium hydroxide (KOH, AR), hydrochloric acid (HCl, AR, 36.0–38.0%), sulfuric acid ($H_2SO_4$, AR, 98%) oxalic acid dihydrate ($H_2C_2O_4 \cdot 2H_2O$, AR), sodium borohydride ($NaBH_4$, AR), acetone ($C_3H_6O$, AR) and ethanol ($C_2H_5OH$, AR) were purchased from Sinopharm Chemical Reagent Co., Ltd. Ruthenium (III) chloride anhydrous ($RuCl_3$) were purchased from Tianjin Xiensi Biochemical Technology Co., Ltd. Ru/C (5 wt%), Pt/C (20 wt%), iridium oxide ($IrO_2$, 99.9%) and Nafion 117 solution (5 wt%) were purchased from Sigma-Aldrich. Ir/C (5 wt%) was purchased from Meryer Chemical Technology Co.,Ltd. Cu foam (CF, thickness: 1.6 mm) was purchased from Shanghai Tankii Alloy Material Co., Ltd.

## Synthesis of $WO_3$ NN

A piece of CF (2 × 4 cm) was etched in HCl solution (1 M) for ~10 min to eliminate the surface oxide layer. Then, it was washed successively with acetone, ultra-pure water and ethanol under ultrasonic condition. $WO_3$ nanoneedle (NN) arrays were grown on CF by a simple hydrothermal process. $Na_2WO_4 \cdot 2H_2O$ (3 mmol) and $H_2C_2O_4$ (8 mmol) were dissolved in 30 mL of ultra-pure water with 250 μL HCl (36.0-38.0%). Subsequently, $(NH_4)_2SO_4$ (6 mmol) and $Na_2SO_4$ (6 mmol) were added into the solution under continuous magnetic stirring for 20 min. The resultant solution was transferred into a 50 mL autoclave and the pre-treated CF was immersed in the solution. The autoclave was heated to 180 °C and maintained at this temperature for 16 h. The obtained $WO_3$ NN arrays on CF were washed with deionized water and ethanol for several times, and dried at 80 °C for 10 h.

## Synthesis of Ru-$H_xWO_3$ NN

$RuCl_3$ (50.0 mg) was dissolved in 20.0 mL of 0.1 M HCl solution through magnetic stirring combined with ultrasonication to prepare the $RuCl_3$ solution (2.5 mg mL$^{-1}$). Subsequently, $WO_3$ NN on CF was immersed in the $RuCl_3$ solution for 1 h. After that, it was washed with deionized water and ethanol for several times, and dried at 80 °C for 2 h. The product was placed into a tubular furnace and heated to 400 °C for 2 h under $H_2$ atmosphere to obtain Ru-$H_xWO_3$ NN. $H_xWO_3$ NN on CF was obtained by following the above procedure while omitting immersing in the $RuCl_3$ solution. The mass loading of Ru-$H_xWO_3$ on CF was controlled to be ~8 mg cm$^{-2}$. The Ir-$H_xWO_3$ and Pt-$H_xWO_3$ NN were prepared by a similar process, with the exception that $RuCl_3$ was replaced by $IrCl_3$ and $H_2PtCl_6$ solution, respectively.

## Synthesis of Ru-$WO_3$ NN

$NaBH_4$ (160.0 mg) was dissolved in 20.0 mL of 1 M NaOH solution to prepare the $NaBH_4$ solution (8.0 mg mL$^{-1}$). Subsequently, $WO_3$ NN on CF was immersed in the $RuCl_3$ solution for 1 h. After that, it was washed with deionized water and ethanol for several times, and dried at 80 °C for 2 h. The resultant product was immersed in the $NaBH_4$ solution to reduce Ru, named as Ru-$WO_3$ NN. The mass loading of Ru-$WO_3$ on CF was controlled to be ~8 mg cm$^{-2}$. The Ir-$WO_3$ and Pt-$WO_3$ NN were prepared by a similar process, with the exception that $RuCl_3$ was replaced by $IrCl_3$ and $H_2PtCl_6$ solution, respectively.

## Preparation of Pt/C and Ru/C electrodes

To prepare the Pt/C or Ru/C electrodes, 32 mg commercial Pt/C or Ru/C, 100 μL Nafion and 900 μL ethanol were ultrasonicated for 60 min to obtain a homogeneous ink. Then, the ink was coated onto a piece of cleaned CF (2 × 2 cm), followed by drying at 80 °C for 10 h. The mass loading of Pt/C or Ru/C on CF was controlled to be ~8 mg cm$^{-2}$.

## Preparation of NiFe-LDH electrodes

The NiFe-layered double hydroxide (NiFe-LDH) was synthesized by a three-electrode system, with the CF (3 × 2 cm), a graphite rod and a saturated calomel electrode (SCE) as the working, counter and reference electrodes, respectively, and 100 mL of solution containing

0.015 mol $Ni(NO_3)_2 \cdot 6H_2O$ and 0.015 mol $FeSO_4 \cdot 7H_2O$ as electrolyte. The electrodeposition was carried out under the $N_2$ atmosphere at a constant potential of -1.0 V vs. SCE for 600 s.

## Materials characterization

X-ray diffraction (XRD) patterns were recorded on a Bruker D2 Advance A25 using a Cu $K_\alpha$ radiation of 1.5406 Å at 30 kV and 10 mA. The morphology and structure were characterized by scanning electron microscope (SEM, Hitachi, S-8100), high-resolution transmission electron microscope (HRTEM, JEM-2100) and high-angle annular dark-field scanning transmission electron microscopy (HAADF-STEM, Thermo Fisher Scientific, FEI Titan Themis 60–300) equipped with energy dispersive X-ray spectrometer (EDS, SUPER X). The Raman spectra were recorded by a confocal Raman spectrometer (Horiba JY Raman) with a laser of 473 nm. $^1$H solid-state nuclear magnetic resonance (NMR) measurements were performed at room temperature on a Bruker 400 MHz NMR spectrometer. Mass spectrometry was recorded on NETZSCH STA 449F3-QMS 403 C Aëolos. X-ray absorption fine structure (XAFS) measurements were carried out on the BL11B beamline of the Shanghai Synchrotron Radiation Facility (SSRF). The obtained XAFS data were analyzed and fitted by using Athena and Artemis software. The surface chemical states and the element composition were analyzed by X-ray photoelectron spectroscopy (XPS, ULVAC-PHI INC, PHI 5000 VersaProbe) with an Al X-ray source, and XPS spectra were calibrated with the C 1$s$ peak at 284.8 eV.

## Thermogravimetric analysis

The $WO_3$ NN powder ( ~ 50 mg) with a little Cu scraped from CF was placed in a crucible and heated to 400 °C in $H_2$/Ar (10%$H_2$/90%Ar) or pure Ar atmosphere with a heating rate of 5 °C min$^{-1}$ for 2 h. The thermogravimetric analysis (TGA) and derivative thermogravimetry (DTG) curves were recorded by NETZSCH STA 449F3 instrument.

## Electrochemical measurements

The electrochemical measurements were conducted on Biologic VMP3 electrochemical workstation in a three-electrode system, with a graphite rod with a diameter of 8 mm as counter electrode and the catalysts on Cu foam (1 × 1 cm) as the working electrodes. The Hg/HgO, SCE, and Hg/$Hg_2SO_4$ electrode were chosen as reference electrodes in 1 M KOH, 1 M phosphate buffered solution (PBS), and 0.5 M $H_2SO_4$, respectively. The 1 M PBS (pH=7.0) was prepared by mixing 1 M $K_2HPO_4$ with 1 M $KH_2PO_4$ in a volume ratio of 2:1. The working electrodes were initially activated through cyclic voltammetry (CV). The CV was performed at a scan rate of 50 mV·s$^{-1}$ for 30 cycles within a potential range from -0.1 to 0.1 V versus reversible hydrogen electrode (vs. RHE). The hydrogen evolution reaction (HER) polarization curves were measured in $N_2$-saturated solution via linear sweep voltammetry (LSV) at a sweep rate of 2 mV s$^{-1}$. To measure the double-layer capacitances ($C_{dl}$), CV was carried out at different scan rates from 10 to 100 mV s$^{-1}$. Subsequently, the electrochemical active surface area (ECSA) could be calculated by the formula ECSA = $C_{dl}/C_s$, where $C_s$ is the specific capacitance with an average value of 0.040 mF cm$^{-2}$. Electrochemical impedance spectroscopy (EIS) was conducted within a frequency range from 0.1 Hz to 100 kHz with an amplitude of 5 mV. The long-term durability was evaluated through chronoamperometry (CP) measurements at a constant current density and room temperature, and the electrolyte was replaced every 48 h. All the measured potentials versus Hg/HgO electrode, SCE, and Hg/$Hg_2SO_4$ electrode were converted to potentials versus RHE according to the equation of $E_{RHE} = E_{Hg/HgO} + 0.059 \times pH + 0.098$, $E_{RHE} = E_{SCE} + 0.059 \times pH + 0.241$, $E_{RHE} = E_{Hg/Hg2SO4} + 0.059 \times pH + 0.653$, respectively. All the polarization and CP curves in this study were corrected for $iR_s$ compensation, and the $iR$ formula is $E_{iR\text{-correction}} = E_{test} - iR_s$, where $R_s$ was measured by EIS tests at 0 V vs. reference electrodes in different electrolytes. To get the activity of the catalyst per unit mass at the overpotential of 100 mV, the

mass activities are calculated by the formula mass activities = $j_{100mV}$/m(Matal), where j represents the current density at the overpotential of 100 mV and m represents the mass of the precious metal.

## Electrochemical Measurements in AEMWE

The membrane electrode assembly (MEA) for the anion exchange membrane water electrolyzer (AEMWE) consisted of a Ru-$H_xWO_3$ NN cathode, a NiFe-LDH anode and a commercial AEM membrane (FAA-3-PK-130). The active area of electrode was 2 cm×2 cm. The AEMWE device was operated at 80 °C with 1 M KOH or 1 M KOH + 0.5 M NaCl as the electrolytes under a flow rate of 20 mL min$^{-1}$. Before the test, AEMWE was activated for 1 h at a current of 0.5 A. The polarization curves were recorded by changing the current density from 0.01 to 1 A cm$^{-2}$. The overall AEMWE device showed the resistance (R) of -0.25 Ω in 1 M KOH and -0.2 Ω in 1 M KOH + 0.5 M NaCl. The polarization curves were corrected by *iR* compensation.

## Measurement of pH on the catalyst surface

The pH values on the catalyst surface were measured via the rotating ring-disk electrode (RRDE) technique, which was carried out on a CHI 760E electrochemical workstation (Shanghai CHI Instruments Company)[26]. The potential of Pt ring electrode (RE, 0.1866 cm$^2$) is sensitive to pH and can be used to monitor the variations in the pH on the surface of disk electrode (DE, 0.2475 cm$^2$) surface. The RRDE and graphite rod were used as working electrode and counter electrode, respectively. The Hg/HgO, SCE, and Hg/Hg$_2$SO$_4$ electrodes were chosen as reference electrodes for H$_2$SO$_4$, PBS and KOH solutions, respectively. Then, the pH dependence of open circuit potential (OCP) was measured with Pt RE in H$_2$-saturated H$_2$SO$_4$, PBS and KOH solutions. The pH of H$_2$SO$_4$ and KOH solution was adjusted by adding 0.5 M H$_2$SO$_4$ or 1 M KOH to 0.5 M K$_2$SO$_4$. The pH of PBS solution was adjusted by adding 1 M H$_3$PO$_4$ or 1.5 M KOH to 1 M PBS. The OCP of the Pt electrode represents the equilibrium potential of the reaction 2H$^+$ + 2e$^-$ → H$_2$, which varies with pH according to the Nernst equation:

$$E_{Rocp}(V \text{ vs. reference electrode}) = \frac{-2.303RT}{F} \text{pH} \tag{1}$$

The fugacity of H$_2$ is assumed to be 1 and $R$, $T$, and $F$ are the gas constant, the absolute temperature, and the Faraday constant, respectively.

To measure the pH on the electrode surface, the catalyst was loaded onto the disk electrode. The catalyst ink was prepared by ultrasonically dispersing catalyst powder (25 mg) scraped from CF in a mixed solution of 5 wt% Nafion solution (80 μL), ethanol (200 μL) and ultra-pure water (720 μL). Then, 10 μL of catalyst ink was dropped onto the disk electrode with catalyst loading of 1 mg cm$^{-2}$. A constant potential method was applied to the disk electrode ($E$ = 0.1, 0, -0.1, -0.2, -0.3, -0.4, -0.5, -0.6, -0.7 V vs. RHE) for 300 s, and the OCP was simultaneously measured on the Pt ring electrode. The pH value of the catalyst-loaded DE can be deducted from the pH value of the Pt RE by the following equation:

$$c_{R,H^+} - c_{R,OH^-} = N_D(c_{D,H^+} - c_{D,OH^-}) + (1 - N_D)(c_{\infty,H^+} - c_{\infty,OH^-}) \tag{2}$$

where $c_{R,H^+}$ and $c_{D,H^+}$ are the concentrations of H$^+$ on the RE and DE, respectively; $c_{R,OH^-}$ and $c_{D,OH^-}$ are the concentrations of OH$^-$ on the RE and DE, respectively; $c_{\infty,H^+}$ and $c_{\infty,OH^-}$ are the concentrations of H$^+$ and OH$^-$ in the bulk electrolyte, respectively. $N_D$ = 0.37 is the collection efficiency of the RE.

## In situ Raman spectroscopy measurements

In situ Raman spectra were recorded by a confocal Raman spectrometer (Horiba JY Raman, with a laser of 473 nm from an argon-ion laser) under specific potentials controlled by CHI 760E electrochemical workstation.

A Pt plate and a SCE electrode were employed as the counter and reference electrode, respectively. The catalysts on CF were directly used as the working electrode. CP measurements were carried out within a specific potential range with the interval of 50 mV in different electrolytes (0.5 M H$_2$SO$_4$, 1 M PBS, and 1 M KOH). Raman tests were initiated once the CP test had exceeded 30 s, and the acquisition time was set to 45 s. For time-resolved in situ Raman spectroscopy, the 1 M PBS and 1 M KOH dissolve in D$_2$O solvent and CP measurements were conducted at constant potential of -0.1 V vs. RHE.

## DFT calculation

Density functional theory (DFT) calculations with the plane-wave basis set were performed using the Vienna Ab Initio Simulation Package (VASP)[35]. The exchange-correlation functional was the Revised Perdew-Burke-Emzerhof (RPBE)[36] of parametrization of the generalized gradient approximation. The electron-ion interactions were described by the projector augmented wave (PAW). The van der Waals interactions were corrected by the DFT-D3 method with Becke-Johnson damping function[37]. A plane-wave basis was set with the cutoff energy of 450 eV. A Gamma-centered 3 × 3 × 1 k-point mesh for Brillouin zone integration was used for static calculations and a 1 × 1 × 1 mesh for structure optimizations. A 2 × 2 × 1 supercell was built from (002) surface of WO$_3$ with a 23.5 Å vacuum along the z-axis and the lattice parameter as 14.8 × 14.8 × 35.0 Å. H$_x$WO$_3$ and Ru-H$_x$WO$_3$ models were constructed base on this slab. VASPKIT was used to assist various computational tasks[38]. Geometry optimization was performed until the convergence criteria were satisfied, with force converging to less than 0.04 eV Å$^{-1}$ and energy difference between ionic steps of less than 10$^{-6}$ eV. VASPsol package was used for counting the implicit solvent effects. The XAFS calculations were carried out with FDMNES code[39]. Transition state searches were conducted using the climbing image nudged elastic band (CI-NEB) method[40], as implemented in the VTST (VASP Transition State Tools) package[41]. The Gibbs free energy of each reaction state was calculated by:

$$G = E_{DFT} + E_{ZPE} - TS \tag{3}$$

The adsorption Gibbs free energy was calculated by:

$$\Delta G_{adsorption} = G_{slab*H} - G_{slab} - 1/2G_{H2} + |e|U_{SHE} + 0.0592 \times pH \tag{4}$$

## Data availability

All data supporting the findings of this study can be found in the main text and the Supplementary Materials, or are available from the corresponding authors upon request. The source data generated in this study are provided in the Source Data file and Supplementary Data 1. Source data are provided with this paper.

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

## Acknowledgements

We thank the BL11B beamline of the Shanghai Synchrotron Radiation Facility (SSRF) for XAFS measurements. The numerical calculations have been done on the computing facilities in the High Performance Computing Center (HPCC) of Nanjing University. This work was jointly supported by the National Key Research and Development Program of China grant 2021YFA1500900 (Z.H.), the National Natural Science Foundation of China grant 52071174 (Z.H.), the National Natural Science Foundation of China grant 22479073 (Q.W.), Major Science and Technology Project of Jiangsu Province grant BG2024033 (Q.W.), the National Key Research and Development Program of China grant 2024YFA1208900 (X.W), and the National Natural Science Foundation of China grant 22322902 (H.H.).

## Author contributions

Y.Z., Q.W., H.H., and Z.H. conceived the idea. Y.Z. designed the study and performed the catalysts' syntheses and characterizations, including their morphology, structure and electrochemical performance. B.F. and L.Y. performed the DFT calculations. J.T. carried out the HRTEM and XAS

measurements. J.T. and S.Z. performed the AEM electrolyser measurements. C.Z. performed the TGA measurements. X.X. and L.P. performed [1]H NMR measurements. Y.Z., B.F., Y.C., X.W., Q.W., H.H., and Z.H. analysed and discussed the experimental and computational data. Y.Z., Q.W., H.H., and Z.H. wrote the article. X.W. and L.Y. supported the project. All authors reviewed the paper.

## Competing interests

The authors declare no competing interests.
