## [Transparent Peer Review file · Nature Communications]

Lattice-Hydrogen Cycling Mechanism Enables pH-Universal Hydrogen Evolution at Ampere-Level Current Densities

Corresponding Author: Professor zheng Hu

Version 0:

Reviewer comments:

Reviewer #1

(Remarks to the Author)

Remarks to the Author: Minor revision

This study proposes an innovative lattice hydrogen cycling concept, which decouples H⁺ availability from electrolyte pH while mitigating local pH fluctuations, the critical limitation in pH-universal HER. The catalyst was experimentally designed through integrating the proton-blocking metallic components with non-proton-blocking support materials (Ru-HxWO₃ NN catalyst). Consistently, due to the lattice-hydrogen cycling pathway, Ru-HxWO₃ NN exhibited unprecedented pH-universal performance. It is thus expected that the proposed strategy and associated mechanistic insights will guide the rational design of advanced catalysts in various hydrogen-related electrochemical processes. Overall, the manuscript presents a logically structured investigation and merits consideration for publication after addressing the following concerns.

1. In this study, the samples as-synthesized exhibit nanoneedle morphology. However, nanostructures with sharp geometrical features typically demonstrate tip-enhanced effects, so does this effect affect the performance of the catalyst in this work?
2. Metallic substrates tend to have a greater influence on the catalytic reaction. In order to exclude the influence of the substrate, could the copper foam used in the manuscript be replaced with another substrate, such as carbon paper?
3. How are WO₃, HxWO₃, Ru-WO₃ and Ru-HxWO₃ modelled? Do their configurations, ratios, and coordination numbers consistent with experiment?
4. The authors posited that variations in the d-band center position and Fermi-level occupied states significantly influence HER process. Please add some representative references after this conclusion.
5. In Supplementary Fig. 17a and c, a significant pH mutation (from ~2 to ~12) is observed. This may not be entirely reasonable, and the authors should provide a brief explanation for this phenomenon.
6. The authors considered Ru and O as active sites. Although W sites in WO₃ was not suitable for HER, the author should add corresponding calculation of W sites for comparison to maintain rigor.
7. In Fig. 5c, only two Ru sites were listed, which cannot reflect superior HER activity of interfacial Ru compared to other sites. The authors should add comparisons with other active sites to enhance persuasiveness.
8. The LSV data for HER without iR compensation needs to be added;
9. In Supplementary Fig. 22, the authors implemented intensity normalization of relative peaks to facilitate their observation. However, the current manuscript lacks corresponding description regarding the specific normalization methodology employed. It is recommended that the authors incorporate a concise description to prevent potential ambiguities.
10. The authors proved the general applicability of the proposed strategy. To further strengthen the comparative analysis, the authors should measure LSV curves of commercially available benchmark counterparts like Ir/C and Pt/C catalysts.

Reviewer #2

(Remarks to the Author)

In this manuscript, Zhang et al. present the integration of proton-blocking Ru with thermally hydrogenated HxWO₃, where a dynamic hydrogen reservoir is established, ensuring a hydrogen supply. This catalyst demonstrates pH-universal performance for the hydrogen evolution reaction (HER), achieving overpotentials as 125 mV in acidic, 142 mV in alkaline, and 219 mV in neutral conditions at 1 A cm², along with stability over 500 h. However, this catalyst and the phenomenon is

already reported in Nature Communications (doi.org/10.1038/s41467-022-33007-3). While the manuscript provides a nice explanation of the reaction mechanism, the material characterization lacks clarity in certain aspects. Comments:

1. XRD pattern shows distinct crystal structures for WO_3 and HxWO_3 . To understand hydrogen insertion phenomenon in WO_3 , it is imperative for the authors to include in-situ XRD analysis.
2. Through experimental details, the authors should clarify whether the amount of reversible hydrogen insertion in WO_3 via electrochemical methods remains consistent or varies in comparison to the pristine sample. Additionally, explain the influence of pH on reversible hydrogen insertion in WO_3 , particularly given its critical role in enhancing catalytic activity.
3. In HxWO_3 , the method used to quantify the x value remains unclear. Provide a detailed procedure for this quantification. Additionally, the amount of hydrogen inserted into WO_3 can be accurately determined by applying an electrochemical bias.
4. Microscopic analysis reveals an unexpected reduction in the (002) interplanar distance from 0.385 to 0.377 nm, following hydrogen insertion into the interlayer galleries of WO_3 . This anomalous behavior warrants further investigation.
5. Include standard deviation of the electrocatalytic activity performance to ensure a comprehensive and accurate representation of the data.
6. The fitting of Ru 3p XPS spectra appears inadequate. I suggest refitting the Ru 3p spectra, as presented in Fig. 2b.

Reviewer #3

(Remarks to the Author)

This study presents a novel Ru- HxWO_3 NN catalyst featuring a lattice hydrogen cycling mechanism that effectively overcomes the pH-dependent limitations of conventional HER electrocatalysts. The engineered dynamic hydrogen storage and migration pathways enable pH-universal operation, delivering exceptional HER performance at ampere-level current densities (125 mV in acidic, 142 mV in alkaline, 219 mV in neutral at 1 A cm^{-2}), coupled with remarkable 500 h stability. This innovative design strategy represents a significant advancement in pH-robust electrocatalyst development, establishing a new paradigm for universal HER systems. However, there are still some critical issues that should be clarified before this manuscript can be considered for publication in Nature Communication. Detailed comments are provided below major issues:

1. It is mentioned that the consumed hydrogen is replenished through proton adsorption (acidic) or water dissociation (alkaline/neutral). Therefore, for the whole lattice hydrogen migration process, three migrations steps are involved instead of twice mentioned in the manuscripts, including hydrogen migration from W sites in WO_3 to OBringing-I, to the terminal O, and finally to the interfacial Ru site. Since the W site provides protons and the oxygen site exhibits positive hydrogen adsorption energy, what is the driving force for the first hydrogen migration from the W site to the O site?
2. Lattice hydrogen involvement often leads to stability problems, how to ensure the balance between lattice hydrogen participation and replenishment.
3. The claim that OCP measurements via RRDE quantitatively track local pH lacks justification. A theoretical framework should be provided.
4. In addition to theoretical calculations, relevant experiments should be proved to elucidate that water molecules tend to adsorb at the W site rather than the Ru site.
5. Given the pivotal role of lattice hydrogen doping in theoretical modes, lattice hydrogen incorporation was quantitatively characterized via XRD should be performed to corroborate TGA-derived doping concentrations and ensure structure accuracy.
6. The current lattice spacing analysis (Figure S5) only examines two selected locations in Ru- WO_3 . To conclusively demonstrate that NaBH_4 reduction does not introduce lattice hydrogen into WO_3 , multiple spatially resolved measurements.
7. Whether relative to WO_3 NN or HxWO_3 , the binding energy of Ru- HxWO_3 moves towards higher binding energy rather than lower binding energy, which needs to be corrected.
8. In EXAFS, the absence of Ru-W bonds cannot be excluded. A RuW alloy reference should be synthesized and measured to validate the above results.
9. In kinetic experiments, the selected potential should be noted in the calculation of the kinetic values in Figure S21e and f.
10. The calculation method of the normalized intensity value of the Raman spectra needs to be described in detail.

Reviewer #4

(Remarks to the Author)

The manuscript by Zhang et al. proposes a novel concept based on the lattice hydrogen cycling mechanism for designing advanced electrocatalysts capable of achieving pH-universal hydrogen evolution reaction (HER) performance. Experimentally, this class of catalyst is realized by integrating a proton-blocking metal with a non-proton-blocking support. The resulting Ru- HxWO_3 nanoneedle (NN) catalyst exhibits outstanding HER activity across a wide pH range, representing a state-of-the-art benchmark for pH-universal HER.

Achieving high HER performance at industrial current densities across varying pH conditions is crucial for the development of sustainable energy conversion technologies. This work addresses a key bottleneck and introduces a transformative strategy with the potential for broad applicability in renewable energy-powered processes. I recommend publication of the manuscript in Nature Communications. The following comments are provided to improve the manuscript:

1. The Ru-WO₃ NN sample without lattice hydrogen was prepared via NaBH₄ treatment. However, NaBH₄ is typically considered a reducing agent that could introduce hydrogen into the lattice. Please clarify why NaBH₄ treatment in this case does not lead to lattice hydrogen insertion into WO₃.
2. The LSV curves presented in Figure 3 lack resolution in the low current density region. Please provide magnified plots of the low-current region to clearly demonstrate the intrinsic catalytic activity.
3. The Tafel slopes of Ru-HxWO₃ NN and the commercial Pt/C catalyst are reported to be approximately 30 mV dec⁻¹ in 0.5 M H₂SO₄ (Figure 3d and Supplementary Figure 14). However, deviations from this value are observed in 1 M PBS and 1 M KOH. Please provide a mechanistic explanation for these deviations.
4. The manuscript reports the mass activities of the catalysts but does not detail the method used for these calculations. Please include a description of the mass activity calculation procedure in an appropriate section of the manuscript.
5. The RRDE-based measurements of local pH reveal significant variations with applied potential, particularly in acidic and alkaline electrolytes. Please add a detailed explanation of the observed strong changes in local pH profiles under these conditions.

Reviewer #5

(Remarks to the Author)

Version 1:

Reviewer comments:

Reviewer #1

(Remarks to the Author)

The revised manuscript has been significantly improved and is now ready for publication in NC.

Reviewer #2

(Remarks to the Author)

The revised manuscript is acceptable.

Reviewer #3

(Remarks to the Author)

In this manuscript, the authors presented a novel Ru-HxWO₃ NN HER catalyst a lattice-hydrogen cycling mechanism that dissociates hydrogen intermediate availability from electrolyte pH. Although the authors have revised the manuscript, several critical issues remain, which should be addressed before publication. The detailed comments are listed as follows.

1. The synthesis method of electrocatalysts in this work is the same as that in the literature (DOI: 10.1038/s41467-022-33007-3), but the explanation is somewhat different. The XRD shift to higher angles is due to the introduction of oxygen vacancies rather than lattice hydrogen. The insertion of lattice hydrogen usually causes lattice expansion, which should lead to a shift towards a lower angle in the XRD pattern. This is consistent with the conclusion stated in the literature (DOI: 10.1038/s41467-023-39963-8). Moreover, the calculation of the amount of inserted hydrogen by the electrochemical method is not accurate. Since HxWO₃ already contains lattice hydrogen, the amount of electrochemically inserted hydrogen should be lower. So, related discussion should be provided.
2. Regarding the calculation of the amount of inserted hydrogen by the electrochemical method, there are a few questions.
 - a. Qa should be the integral of current over time. If current density is used, it should be multiplied by the electrode area.
 - b. According to the supporting information in reference 9, this method is not accurate for quantification, as it takes into account the current generated by the hydrogen evolution reaction, leading to a value lower than the actual one.
 - c. In supplementary Fig. 3h, why is the calculated hydrogen content in HxWO₃ higher than that in WO₃? Since HxWO₃ already contains lattice hydrogen, shouldn't the amount of electrochemically inserted hydrogen be lower?
3. For DFT calculations, how is the HxWO₃ model constructed, and how is the hydrogen content (x) determined? Based on the TGA results, the x value in HxWO₃ NN support is about 0.88-0.97. However, in supplementary Fig. 30, is the x value of the HxWO₃ model 0.75? In comment 3 for reviewer 1, the x for the Ru-HxWO₃ model is 0.625. What is the basis for this value?
4. The synthesis method used in this work is the same as that in the literature (DOI: 10.1038/s41467-022-33007-3), but the explanation for the XRD peak shift toward higher angles is completely different. The reference attributes it to the presence of oxygen vacancies, rather than hydrogen insertion. According to another literature (DOI: 10.1038/s41467-023-39963-8), the insertion of hydrogen leads to a shift of the XRD diffraction peaks of HxWO₃ toward lower angles in supplementary Fig. 10.

How should this be understood?

5. According to the literature (DOI: 10.1038/s41467-022-33007-3), with an almost identical synthesis method, the oxygen vacancies and reversible hydrogen spillover play a crucial role in enhancing the HER activity. This work did not take into account the influence of oxygen vacancies.

6. In supplementary Fig. 21, the HER performance of Ru-HxWO₃ is better than that of Ru-WO₃, which cannot be simply attributed to the effect of lattice hydrogen. The two materials were synthesized using completely different methods: the former was reduced by H₂ annealing, while the latter was reduced by NaBH₄. It may introduce the B element.

7. In supplementary Fig. 20c, the pH value in the KOH solution changed from 13 to 2, which is indeed a bit strange.

8. In Figure 4d-f, the decrease in the Raman signals of O-W-O indicates the insertion of hydrogen. Meanwhile, the signals of W-O-H should increase. Only HxWO₃ and Ru-WO₃ conform to this phenomenon. However, for Ru-HxWO₃, the Raman signal of WO-H decreases, which is quite strange.

Reviewer #4

(Remarks to the Author)

The manuscript can be accepted in the present form.

Reviewer #5

(Remarks to the Author)

Version 2:

Reviewer comments:

Reviewer #3

(Remarks to the Author)

The authors have revised the manuscript and it is recommended to be accepted.

The point-by-point response to the reviewers' comments for NCOMMS-25-25972-T
(For clarity, the reviewers' comments are quoted in *italics* ahead of the corresponding responses.)

Reviewer #1:

Remarks to the Author: Minor revision

This study proposes an innovative lattice hydrogen cycling concept, which decouples H^ availability from electrolyte pH while mitigating local pH fluctuations, the critical limitation in pH-universal HER. The catalyst was experimentally designed through integrating the proton-blocking metallic components with non-proton-blocking support materials (Ru- H_x WO₃ NN catalyst). Consistently, due to the lattice-hydrogen cycling pathway, Ru- H_x WO₃ NN exhibited unprecedented pH-universal performance. It is thus expected that the proposed strategy and associated mechanistic insights will guide the rational design of advanced catalysts in various hydrogen-related electrochemical processes. Overall, the manuscript presents a logically structured investigation and merits consideration for publication after addressing the following concerns.*

General Response: Thanks for your encouraging comments.

Comment 1. *In this study, the samples as-synthesized exhibit nanoneedle morphology. However, nanostructures with sharp geometrical features typically demonstrate tip-enhanced effects, so does this effect affect the performance of the catalyst in this work?*

Response to Comment 1: Thanks for your valuable question.

In order to reveal the tip-enhanced effects of Ru- H_x WO₃ NN, we have prepared WO₃ nanoarrays with different morphologies by adjusting the type of cations in the hydrothermal synthesis. It is found that the WO₃ nanowire (NW) array was prepared when only (NH₄)₂SO₄ was employed and the WO₃ nanorod (NR) array was obtained when only Na₂SO₄ was used. The corresponding Ru- H_x WO₃ NW and Ru- H_x WO₃ NR show the same crystal structure and composition but different tip morphology, as depicted in **Figure R1**. The LSV curves indicate that Ru- H_x WO₃ NN exhibits the better performances in 0.5 M H₂SO₄, 1M PBS and 1M KOH solution than Ru- H_x WO₃ NW and Ru- H_x WO₃ NR (**Figure R2**). The results of this comparative experiment show the marked tip-enhanced effect of the Ru- H_x WO₃ NN on the HER performance.

Due to limitation in length and the focus of the content, this point was not included in the manuscript of this study. We will explore this phenomenon separately in our next work.

Figure R1 (a-c) SEM (a), TEM (b) and HRTEM (c) images of Ru- H_x WO₃ NW; (d-f) SEM (d), TEM (e) and HRTEM (f) images of Ru- H_x WO₃ NR. Insets in (c) and (f) are the corresponding local enlargements; (g) XRD patterns and the corresponding local enlargements in 20-30°; (h) Raman spectra; (i) ¹H NMR spectra. The samples are marked in the figures, and the spectra for Ru- H_x WO₃ NN are also shown for comparison.

Figure R2 LSV curves of Ru- H_x WO₃ NN, NW and NR in 0.5 M H₂SO₄ (a), 1 M PBS (b) and 1 M KOH (c).

Comment 2. *Metallic substrates tend to have a greater influence on the catalytic reaction. In order to exclude the influence of the substrate, could the copper foam used in the manuscript be replaced with another substrate, such as carbon paper?*

Response to Comment 2: Thanks for your thoughtful suggestion.

According to previous reports (Dopov. Akad. Nauk Ukr. RSR, Ser. B, 1985, 1, 46-49; J. Mater. Sci., 1988, 23, 267; J. Alloys Compd., 2001, 320, 1-6), the presence of Cu elements is beneficial for the hydrogen insertion into WO₃. We replaced the Cu foam substrate with carbon paper (CP) and

carbon cloth (CC) during the synthesis. From the XRD patterns, it is found that, for the WO_3 nanoarrays deposited on CP and CC ($\text{WO}_3@CP$, $\text{WO}_3@CC$), no significant shifts were observed for the (002) peak after the H_2 calcination (**Figure R3**), different from the samples grown on Cu foam. This result indicates that the Cu substrate can facilitate the H insertion into the WO_3 nanoarrays.

Figure R3 XRD patterns of WO_3 on CP and CC before and after H_2 calcination.

Comment 3. How are WO_3 , H_xWO_3 , Ru-WO_3 and $\text{Ru-H}_x\text{WO}_3$ modelled? Do their configurations, ratios, and coordination numbers consistent with experiment?

Response to Comment 3: Thanks for your thoughtful questions.

Based on the XRD patterns (**Fig. 1e**) and EXAFS results (**Fig. 2c-f**), theoretical models of the above four materials were constructed for DFT calculation, as shown in **Supplementary Fig. 30**. The relevant coordination information was added in Supplementary Fig. 30. Specifically, the WO_3 model was chosen from XRD standard card PDF: # 84-2460. For the convenience of calculation, the Ru species were set as a Ru_{19} cluster and the amount of H per single W was set as 0.625, which are consistent with the experimental results and previous reports (Nat. Commun. 2022, 13, 5382; Nat. Commun. 2023, 14, 4209; Chem. Sci. 2024, 15, 5385-5402).

Supplementary Fig. 30. The side illustrations of models for DFT calculation. **a** WO_3 . **b** H_xWO_3 . **c** Ru-WO_3 . **d** $\text{Ru-H}_x\text{WO}_3$.

Comment 4. The authors posited that variations in the d -band center position and Fermi-level occupied states significantly influence HER process. Please add some representative references after this conclusion.

Response to Comment 4: Thanks for your thoughtful suggestion.

Ref. 23 and 24 have been added at **Page 6 Paragraph 1** as follows:

“This electron redistribution causes the d -band centers of Ru and W in the $\text{Ru-H}_x\text{WO}_3$ system to shift closer to the Fermi level (E_f) compared to those in Ru-WO_3 , which can enhance the adsorption stability of key intermediates (e.g., H^* , H_2O^*) on $\text{Ru-H}_x\text{WO}_3$ ²³(Ref. 23). Moreover, the higher Fermi level occupancy ($E-E_f=0$) for Ru and W in $\text{Ru-H}_x\text{WO}_3$ further facilitates the electron conductivity and HER thereof (Fig. 2h,i)²⁴(Ref. 24).”

Comment 5. In Supplementary Fig. 17a and c, a significant pH mutation (from ~ 2 to ~ 12) is observed. This may not be entirely reasonable, and the authors should provide a brief explanation for this phenomenon.

Response to Comment 5: Thanks for your thoughtful suggestion.

As described in **Supplementary Materials** (Page 3), the OCP of the Pt-ring electrode ($E_{R,\text{ocp}}$) represents the equilibrium potential of the reaction $2\text{H}^+ + 2\text{e}^- \rightarrow \text{H}_2$, which varies with the local pH near the Pt-ring electrode (RE) according to the Nernst equation:

$$E_{R,\text{ocp}}(\text{V vs. reference electrode}) = \frac{-2.303RT}{F} \text{pH}_R \quad (1)$$

The pH_D value of the catalyst-loaded disk electrode (DE) can be deduced from the pH_R value by the following equation:

$$c_{R,\text{H}^+} - c_{R,\text{OH}^-} = N_D(c_{D,\text{H}^+} - c_{D,\text{OH}^-}) + (1 - N_D)(c_{\infty,\text{H}^+} - c_{\infty,\text{OH}^-}) \quad (2)$$

where, N_D is the collection efficiency at the RE ($N_D = 0.37$), c_{∞,H^+} and c_{∞,OH^-} are the concentrations of H^+ and OH^- in the bulk electrolyte, respectively. It is seen that the $E_{R,\text{ocp}}$ can be used to deduce the pH_R (Equation 1) and further calculate the pH_D on the DE (Equation 2), and the c_R terms in Equation 2 are all obtained from the OCP testing. There is the specific $E_{R,\text{ocp}}$ corresponding to $\text{pH}_D = 7$ during testing. Due to the significant impact of the last term in Equation

2, which is large in strong alkaline and acidic solutions, ultrasmall disturbances near the above specific $E_{R,ocp}$ can cause drastic changes of pH_D , which affects testing near this potential. However, this does not affect the trend over a large potential range.

A brief description about this point is provided in **Supplementary Fig. 20**:

“**Note:** It should be pointed out that this pH measurement method is more accurate under neutral conditions ¹. Due to the relatively large value of c_{∞,H^+} or c_{∞,OH^-} in Equation (2) in strongly acidic or alkaline environments, there is the specific OCP of Pt-RE corresponding to $pH=7$ on DE during testing. Ultrasmall disturbances near the above specific OCP of Pt-RE can cause drastic changes in pH on DE, but this does not affect the trend over a large potential range and the above conclusion was verified in this work.”

Comment 6. *The authors considered Ru and O as active sites. Although W sites in WO_3 was not suitable for HER, the author should add corresponding calculation of W sites for comparison to maintain rigor.*

Comment 7. *In Fig. 5c, only two Ru sites were listed, which cannot reflect superior HER activity of interfacial Ru compared to other sites. The authors should add comparisons with other active sites to enhance persuasiveness.*

Response to Comment 6 and 7: Thanks for your thoughtful suggestions.

Generally, H atom cannot be adsorbed on the coordination-saturated W sites, and the coupling of two H atoms is very difficult due to the too large distance between adjacent W sites (Nat. Commun., 2023, 14, 4209). According to previous report (J. Am. Chem. Soc. 2022, 144, 6420–6433), the W sites with oxygen vacancy ($W_{O-vacancy}$) could act as active sites for HER. Therefore, the free energies for H adsorbed on single $W_{O-vacancy}$ site and the coupling of two H on adjacent $W_{O-vacancy}$ sites were calculated in order to exhibit the HER activity of W sites. The calculation results were added as **Supplementary Fig. 32**, which show the adsorption free energy of H on the W sites with O vacancy ($W_{O-vacancy}$).

In addition, we added the free energy profiles of HER on $Ru_{interfacial}$ in WO_3 and Ru_{top} in WO_3 into **Fig. 5c**, confirming the excellent HER capability of the interfacial Ru in $Ru-H_xWO_3$.

The corresponding description was exchanged to **Page 10, Paragraph 1**:

“Moreover, the ΔG_{H^*} for the $Ru_{interfacial}$ in H_xWO_3 is -0.08 eV, very close to the ideal value for HER. In contrast, the ΔG_{H^*} for the possible Ru and W active sites remain highly positive, showing the inferior HER activity of these sites (Fig. 5b,c, Supplementary Fig. 32)³⁴.”

Fig. 5c Corresponding free energy profiles for HER on different Ru sites.

Supplementary Fig. 32 Calculation for H adsorbed on W sites. **a** WO_3 . **b** H_xWO_3 . **c** Corresponding free energy profiles for HER on single W sites. **d** Corresponding free energy profiles for HER on adjacent W sites.

Comment 8. The LSV data for HER without iR compensation needs to be added;

Response to Comment 8: Thanks for your thoughtful suggestion.

The LSV curves without iR -compensation were added in **Supplementary Fig. 9d-f**. Fig. 3a-c are the compensated results of these LSV curves by the formula of $E_{iR\text{-compensation}} = E_{\text{test}} - iR_s$ (R_s was measured by EIS tests at 0 V vs. reference electrodes in different electrolytes).

Supplementary Fig. 9. Polarization curves. **d-f** The LSV curves without iR -compensation in 0.5 M H_2SO_4 (d), 1 M PBS (e) and 1 M KOH (f).

Comment 9. In Supplementary Fig. 22, the authors implemented intensity normalization of relative peaks to facilitate their observation. However, the current manuscript lacks corresponding description regarding the specific normalization methodology employed. It is recommended that the authors incorporate a concise description to prevent potential ambiguities.

Response to Comment 9: Thanks for your valuable suggestion.

The specific normalization method is performed as follows: using the peak intensity at 0 V (vs. RHE) as the reference value, and then calculate the ratio of the peak intensity at other potentials to the reference value.

The corresponding description was added in **Supplementary Fig. 25**:

“The normalization method: the peak intensity at 0 V (vs. RHE) was set as the reference value, and the normalized intensity at a certain potential is the ratio of the corresponding peak intensity to the reference value.”

Comment 10. The authors proved the general applicability of the proposed strategy. To further strengthen the comparative analysis, the authors should measure LSV curves of commercially available benchmark counterparts like Ir/C and Pt/C catalysts.

Response to Comment 10: Thanks for your thoughtful suggestion.

We have added the performance of Pt/C and Ir/C benchmark catalysts in **Supplementary Fig. 34** for comparison. Note: Pt/C (20 wt%) was purchased from Sigma-Aldrich, and Ir/C (5 wt%) was purchased from Meryer Chemical Technology Co., Ltd. The LSV curves for benchmark catalysts are also corrected by iR -compensation.

Supplementary Fig. 34. Polarization curves. a-c LSV curves of Ir-H_xWO₃ NN, Ir-WO₃ NN and Ir/C in 0.5 M H₂SO₄ (a), 1 M PBS (b) and 1 M KOH (c). d-f LSV curves of Pt-H_xWO₃ NN, Pt-WO₃ NN and Pt/C in 0.5 M H₂SO₄ (d), 1 M PBS (e) and 1 M KOH (f).

Reviewer #2:

In this manuscript, Zhang *et al.* present the integration of proton-blocking Ru with thermally hydrogenated H_xWO₃, where a dynamic hydrogen reservoir is established, ensuring a hydrogen supply. This catalyst demonstrates pH-universal performance for the hydrogen evolution reaction (HER), achieving overpotentials as 125 mV in acidic, 142 mV in alkaline, and 219 mV in neutral

conditions at 1 A cm², along with stability over 500 h. However, this catalyst and the phenomenon is already reported in Nature Communications (doi.org/10.1038/s41467-022-33007-3). While the manuscript provides a nice explanation of the reaction mechanism, the material characterization lacks clarity in certain aspects. Comments:

General Response: We are very grateful for your positive comments. We have a different opinion on the concerns you raised regarding the catalyst and phenomenon that have been reported (doi.org/10.1038/s41467-022-33007-3, i.e. Nat. Commun. 2022, 13, 5382). Although both catalysts in these two works are the Ru/WO₃ hybrid system, they are actually quite different.

First, in the study you mentioned, the Ru NPs were loaded on the WO_{3-x} nanowires on carbon paper, and the H_xWO_{3-x} was formed during electrochemical hydrogenation. In our study, the H_xWO₃ NN was pre-hydrogenated by thermal H₂ reduction and the lattice H atoms can be further inserted in the electrochemical process. Thus, there are more lattice H atoms (see Response to Comment 1~3 below) in our Ru-H_xWO₃ NN sample to act as “H reservoir”, leading to the enhanced HER performance. It is clear that our Ru-H_xWO₃ NN and the Ru-WO_{3-x} in literature are much different in terms of structure and composition.

Second, this article is completely different in terms of design concept, performance and mechanism. Specifically,

1. **Groundbreaking concept:** We proposed a new concept of lattice-hydrogen cycling for HER: Create a “dynamic H reservoir”, which can rapidly supply the lattice hydrogen to the active sites and the consumed hydrogen can be replenished independent of electrolyte pH. To prove this concept, we integrated the proton-blocking metal Ru as the active sites for pH-universal HER, with non-proton-blocking support H_xWO₃ as the “dynamic H reservoir”. However, the study you mentioned started from addressing the pain points of water electrolysis technology in neutral medium, and the catalyst design is guided by the phenomenon of hydrogen spillover. Therefore, these two articles are different in terms of mechanism.
2. **Record-high pH-universal HER performance:** The as-designed Ru-H_xWO₃ catalyst exhibits an unprecedented pH-universal HER performance with ultralow overpotentials of 125, 219, and 142 mV at an industrial current density of 1 A cm⁻² in respective 0.5 M H₂SO₄, 1 M phosphate buffered solution, and 1 M KOH, as well as the outstanding durability (over 500 h) in all-pH electrolytes. However, the study you mentioned only focused on the performance under neutral conditions, and the durability of catalysts was tested at 20 mA cm⁻² only for 30 h.
3. **Rigorous mechanism validation:** We used *in situ* Raman spectroscopy, isotopic labeling, and density functional theory calculations to verify the rapid lattice-hydrogen migration and replenishment pathways. The results clearly reveal that the lattice-hydrogen in H_xWO₃ migrates swiftly to Ru active sites via low-energy pathways, and meanwhile, the consumed hydrogen is spontaneously replenished through proton adsorption (acidic) or water dissociation (alkaline/neutral). Such a unique mechanism reduces the impact of electrolyte pH value on H* formation, ensuring sustained pH-universal catalytic activity. However, the study you mentioned paid more attention to the influence of oxygen vacancies on WO_{3-x} and the phenomenon of hydrogen spillover, neglecting experimental phenomenon of lattice hydrogen existence and replenishment.

Based on the above reasons, we believe, the “dynamic H reservoir”, pH-Universal HER performance and lattice-hydrogen cycling mechanism of our study is novel and eligible for publication in Nature Communications.

Comment 1. XRD pattern shows distinct crystal structures for WO_3 and H_xWO_3 . To understand hydrogen insertion phenomenon in WO_3 , it is imperative for the authors to include *in-situ* XRD analysis.

Response to Comment 1: Thanks for your thoughtful suggestion.

The WO_3 NN arrays on Cu foam was thermal-hydrogenated in 10% H_2/Ar atmosphere and the XRD patterns were *in situ* recorded on Smartlab SE using a $Cu K_{\alpha}$ radiation of 1.5406 \AA at 40 kV and 40 mA (Note: The 10% H_2/Ar gas was used to replace pure H_2 , for avoiding the damage to the XRD instrument). The temperature was increased at a heating rate of $5 \text{ }^{\circ}C \text{ min}^{-1}$ from room temperature to $400 \text{ }^{\circ}C$ and then cooling down to room temperature, and the *in situ* XRD patterns were recorded with a temperature interval of $50 \text{ }^{\circ}C$ in the range of $100 \text{ }^{\circ}C$ to $400 \text{ }^{\circ}C$.

The *in situ* XRD patterns show that the (002) peak gradually shifts to high angles during the temperature-increasing process, and the position of (002) peak keeps constant in the subsequent cooling process, which confirms the formation of H_xWO_3 during the thermal-hydrogenation.

The *in situ* XRD patterns and corresponding descriptions were supplemented as **Supplementary Fig. 7**, and the result was discussed in the main text at **Page 3, Paragraph 4**:

“...from 23.2° to 23.6° . The *in situ* XRD patterns show that the (002) peak gradually shifts to high angles during the temperature-increasing process, and the position of (002) peak keeps constant in the subsequent cooling process, which confirms the formation of H_xWO_3 during the thermal-hydrogenation (Supplementary Fig. 7). This shift also reflects...”.

Supplementary Fig. 7. *In situ* XRD patterns for WO_3 sample thermal-hydrogenated at different temperatures under 10% H_2/Ar atmosphere. The local enlargements of the patterns in the range of 20° - 30° are shown for clarity.

The WO_3 NN arrays on Cu foam was thermal-hydrogenated in 10% H_2/Ar atmosphere and the XRD patterns were *in situ* recorded on Smartlab SE with a temperature interval of $50 \text{ }^{\circ}C$ in the range of $100 \text{ }^{\circ}C$ to $400 \text{ }^{\circ}C$.

Comment 2. Through experimental details, the authors should clarify whether the amount of reversible hydrogen insertion in WO_3 via electrochemical methods remains consistent or varies in

comparison to the pristine sample. Additionally, explain the influence of pH on reversible hydrogen insertion in WO_3 , particularly given its critical role in enhancing catalytic activity.

Response to Comment 2: Thanks for your thoughtful suggestion.

According to previous report (Chem. Sci. 2024, 15, 5385-5402), when the H insertion reaction occurs in the cathodic CV sweeping, the anodic capacity (Q_a) is often used to quantify the amount of H inserted into a material. Therefore, the CV tests were used to quantify the amount of reversible hydrogen insertion in WO_3 NN and H_xWO_3 NN. The WO_3 NN and H_xWO_3 NN powders were stripped from the Cu foam by ultrasonication, and then coated on carbon paper as working electrode with a mass loading of 1 mg cm^{-2} . The CV scan rate is 10 mV s^{-1} . The integral of the positive current with respect to time for one CV cycle is taken to calculate the Q_a , which was converted to the number of hydron atom per tungsten atom by the following equations:

$$Q_a = \int j dt$$
$$n_H = \frac{Q_a \times N_e}{N}$$
$$x = \frac{n_H}{n_{WO_3}}$$

where j is the positive current, n_H is the amount of substance for H insertion, N_e is the number of electrons per Coulomb; N is the Avogadro constant; n_{WO_3} is the amount of substance for WO_3 .

The CV curves and deduced Q_a and x value in different electrolytes were added as **Supplementary Fig. 3**. It was found that after the H insertion via electrochemical method, the x value of WO_3 NN was 0.45 in 0.5 M H_2SO_4 , similar with the above report; and the x value of H_xWO_3 NN was 0.80 in 0.5 M H_2SO_4 which was obviously higher than that of WO_3 NN. For H_xWO_3 NN, the measured x value was the amount of H that can achieve reversible hydrogen insertion. This value was only slightly lower than the value obtained from TGA (Supplementary Fig. 3).

In addition, the x values of WO_3 NN were much lower when tested in neutral ($x=0.18$) and alkaline ($x=0.26$) electrolytes due to the increased difficulty in obtaining H species, indicating a poor hydrogen supply for WO_3 NN during HER. However, the x values of H_xWO_3 NN were 0.6 in neutral electrolyte and 0.68 in alkaline electrolyte, thanks to its high initial concentration and faster replenishment of lattice hydrogen. Therefore, the Ru- H_xWO_3 NN exhibits much better HER performances than Ru- WO_3 NN in neutral and alkaline electrolytes.

Due to the fact that Ru metal also undergoes electrochemical adsorption/desorption reactions within the range of the CV testing, the x value of Ru- H_xWO_3 NN catalyst will be significantly interfered. Therefore, the x value in Ru- H_xWO_3 NN before and after HER are difficult to be measured.

The result was discussed in the main text at **Page 3, Paragraph 2 and Supplementary Fig. 3**:

“...the WO_3 lattice. CV tests and thermogravimetric analysis (TGA) further confirm the insertion of hydrogen into WO_3 NN, with x value of ~ 0.8 in H_xWO_3 NN (Supplementary Fig. 3 and Fig. 4). It is worth noting that when protons are inserted into WO_3 to form H_yWO_3 by electrochemical method, the typical value of y is ca. 0.5, which is significantly lower than that in the thermally-hydrogenated H_xWO_3 NN (Supplementary Fig. 3)”

Supplementary Fig. 3 CV curves, deduced Q_a and x value in 0.5 M H_2SO_4 , 1 M PBS and 1 M KOH. a-f CV curves. g,h deduced Q_a and x value.

The CV tests were used to quantify the amount of reversible hydrogen insertion in WO_3 NN and H_xWO_3 NN⁹. The WO_3 NN and H_xWO_3 NN powders were stripped from the Cu foam by ultrasonication, and then coated on carbon paper as working electrode with a mass loading of 1 mg cm^{-2} . The CV scan rate is 10 $mV s^{-1}$. The integral of the positive current with respect to time for one CV cycle is taken to calculate the Q_a , which was converted to the number of hydron atom per tungsten atom by the following equations:

$$Q_a = \int j dt$$

$$n_H = \frac{Q_a \times N_e}{N}$$

$$x = \frac{n_H}{n_{WO_3}}$$

where j is the positive current, n_H is the amount of substance for H insertion, N_e is the number of electrons per Coulomb; N is the Avogadro constant; n_{WO_3} is the amount of substance for WO_3 .

It was found that after the H insertion via electrochemical method, the x value of WO_3 NN was 0.45 in 0.5 M H_2SO_4 , similar with the above report; and the x value of H_xWO_3 NN was 0.80 in 0.5 M H_2SO_4 which was obviously higher than that of WO_3 NN. For H_xWO_3 NN, the measured x value was the amount of H that can achieve reversible hydrogen insertion.

In addition, the x values of WO_3 NN were much lower when tested in neutral ($x=0.18$) and alkaline ($x=0.26$) electrolytes due to the increased difficulty in obtaining H species, indicating a poor hydrogen supply for WO_3 NN during HER. However, the x values of H_xWO_3 NN were 0.6 in neutral electrolyte and 0.68 in alkaline electrolyte, thanks to its high initial concentration and faster replenishment of lattice hydrogen, which is contributed to achieving similar HER performance in different electrolyte.

Comment 3. In H_xWO_3 , the method used to quantify the x value remains unclear. Provide a detailed procedure for this quantification. Additionally, the amount of hydrogen inserted into WO_3 can be accurately determined by applying an electrochemical bias.

Response to Comment 3: Thanks for your thoughtful suggestion.

The TGA curves of WO_3 NN were recorded under 10% H_2 /90%Ar and pure Ar atmosphere,

with the temperature ranging from room temperature to 400 °C, which exhibit a continuous downward trend due to the slight loss of crystal water. The DTG curves show that the weight-loss rate under the 10%H₂/90%Ar atmosphere is slower than that under the Ar atmosphere because of the H insertion into WO₃ (Supplementary Fig. 4a).

Regarding the TGA curves of WO₃ NN at 400 °C under the H₂/Ar or Ar atmosphere, the difference of weight loss is approximately 0.38-0.42 % (Supplementary Fig. 4b), which can be used to calculate the value of x in H _{x} WO₃. It is known that there is no change in the amount of substance (n) during TGA testing. We described the calculation method for determining the specific x value in H _{x} WO₃ using the TG data at **Supplementary Fig. 4**:

“It is known that there is no change in the amount of substance (n) for WO₃ during TGA. Therefore, the value of x can be calculated using the following formula:

$$m_H = m_{\text{WO}_3-\text{H}_2} - m_{\text{WO}_3-\text{Ar}} = \Delta\omega \times m_{\text{initial}}$$

$$x \times n \times M_H = \Delta\omega \times n \times M_{\text{WO}_3}$$

$$x = \frac{M_{\text{WO}_3} \times \Delta\omega}{M_H}$$

where M is the molecular weight, $\Delta\omega$ is the weight-loss difference, and m is the mass. Based on the TGA results, the x value in H _{x} WO₃ NN support is about 0.88-0.97.”

In “**Response to Comment 2**”, the CV tests were used to quantify the amount of reversible hydrogen insertion in WO₃ NN and H _{x} WO₃ NN. The x value in H _{x} WO₃ NN was tested as about 0.80 by the CV tests, which is in consistence with the value calculated based on TGA result.

Comment 4. *Microscopic analysis reveals an unexpected reduction in the (002) interplanar distance from 0.385 to 0.377 nm, following hydrogen insertion into the interlayer galleries of WO₃. This anomalous behavior warrants further investigation.*

Response to Comment 4: Thanks for your thoughtful suggestion.

The reduction in the (002) interplanar distance from 0.385 nm for WO₃ to 0.377 nm for H _{x} WO₃ is reasonable. The XRD patterns of our H _{x} WO₃ samples and the standard JCPDF cards for H _{x} WO₃ do show the larger diffraction angles for (002) peaks than that for WO₃ (**Fig. R4**). In fact, the lattice hydrogen atom is bonded to the bridging O atom, rather than directly inserted into the bond of W-O-W, as shown in **Supplementary Fig. 27**. The insertion of lattice hydrogen results in a shortening of bond length and thus a positive shift of the corresponding XRD peak. In addition, the DFT calculation also reveals that the insertion of lattice hydrogen atom can shorten the lattice spacing from 0.411 nm for WO₃ to 0.398 nm for H _{x} WO₃ (**Fig. R5**).

Fig. R4 Standard JCPDF XRD patterns for WO₃ and H _{x} WO₃.

Fig. R5 Lattice spacing of WO_3 and H_xWO_3 by calculation.

Comment 5. Include standard deviation of the electrocatalytic activity performance to ensure a comprehensive and accurate representation of the data.

Response to Comment 5: Thanks for your thoughtful suggestion.

The polarization curves have been repeatedly tested for at least three times, and the error bars for the overpotential at 1 A cm^{-2} were added into **Supplementary Fig. 12** to ensure a comprehensive and accurate representation of the data.

Supplementary Fig. 12. Overpotentials at 1 A cm^{-2} and mass activities at 100 mV . (a) $0.5 \text{ M H}_2\text{SO}_4$; (b) 1 M PBS ; (c) 1 M KOH . The samples are marked in the figures. **Note:** The error bars for the overpotential at 1 A cm^{-2} are obtained from the polarization curves which is repeatedly tested for at least three times.

Comment 6. The fitting of Ru 3p XPS spectra appears inadequate. I suggest refitting the Ru 3p spectra, as presented in Fig. 2b.

Response to Comment 6: Thanks for your thoughtful suggestion.

The whole Ru 3p XPS fine spectra (including $3p_{3/2}$ and $3p_{1/2}$) were refitted and updated to **Fig. 2b**.

Fig. 2b Ru 3p spectra for $\text{Ru-H}_x\text{WO}_3 \text{ NN}$, $\text{Ru-WO}_3 \text{ NN}$ and Ru/C .

Reviewer #3

This study presents a novel Ru-H_xWO₃ NN catalyst featuring a lattice hydrogen cycling mechanism that effectively overcomes the pH-dependent limitations of conventional HER electrocatalysts. The engineered dynamic hydrogen storage and migration pathways enable pH-universal operation, delivering exceptional HER performance at ampere-level current densities (125 mV in acidic, 142 mV in alkaline, 219 mV in neutral at 1 A cm⁻²), coupled with remarkable 500 h stability. This innovative design strategy represents a significant advancement in pH-robust electrocatalyst development, establishing a new paradigm for universal HER systems. However, there are still some critical issues that should be clarified before this manuscript can be considered for publication in Nature Communication. Detailed comments are provided below major issues:

General Response: We truly appreciate your positive comment and valuable questions.

Comment 1. It is mentioned that the consumed hydrogen is replenished through proton adsorption (acidic) or water dissociation (alkaline/neutral). Therefore, for the whole lattice hydrogen migration process, three migration steps are involved instead of twice mentioned in the manuscripts, including hydrogen migration from W sites in WO₃ to O_{Bridging-I}, to the terminal O, and finally to the interfacial Ru site. Since the W site provides protons and the oxygen site exhibits positive hydrogen adsorption energy, what is the driving force for the first hydrogen migration from the W site to the O site?

Response to Comment 1: Thanks for your thoughtful question.

Actually, all the H migrations mentioned in this manuscript are occurred between different O sites. As mentioned in **Supplementary Fig. 28**, the W sites exhibit much high positive free energy of H adsorption (**Please see Response to Comment 6 and 7 of Reviewer #1**). Therefore, it is difficult for W sites to bind with H atom. For the water dissociation on H_xWO₃, the O atom in H₂O usually binds to the W sites, while the H atom in H₂O is strongly adsorbed on the adjacent O sites, thereby achieving the dissociation of H₂O molecules. Then, the H atoms migrate from the terminal O to the interfacial Ru site (Fig. 5a). In order to more clearly illustrate the water dissociation process on H_xWO₃, we have added the diagrams of the H₂O dissociation process in **Supplementary Fig. 31d**.

Supplementary Fig. 31 d The process of H₂O molecule dissociation on W sites in Ru-H_xWO₃.

Comment 2. Lattice hydrogen involvement often leads to stability problems, how to ensure the balance between lattice hydrogen participation and replenishment.

Response to Comment 2: Thanks for your valuable question.

Lattice hydrogen atom is bonded to the O atom; therefore, the migration of H atoms does not break the W-O-W bond. After the migration of lattice H atoms, WO₃ remains its stable crystal structure. Moreover, with the assistance of the W site, the dissociation of water molecules can be promoted, which can quickly replenish H to the O sites (Supplementary Fig. 30d). Therefore, H_xWO₃ is stable during lattice-hydrogen cycling, which is responsible to the high HER stability.

Comment 3. *The claim that OCP measurements via RRDE quantitatively track local pH lacks justification. A theoretical framework should be provided.*

Response to Comment 3: Thanks for your thoughtful suggestion.

The local pH was quantitatively tracked by the OCP measurements via RRDE, which have been widely adopted in many literatures (ChemElectroChem 2019, 6, 4750-4756; Nat. Energy 2023, 8, 264–272; Adv. Mater. 2025, 2507525; J. Am. Chem. Soc. 2025, 147, 20, 17190–17200; Sci. Adv. 2024, 10, eadn7012). The detailed process, along with clear theoretical explanations, is provided in the article that we have cited in the Supplementary Materials as Reference 1 (ChemElectroChem 2019, 6, 4750-4756). The specific theoretical explanations were too long to be inconveniently listed in this article.

Comment 4. *In addition to theoretical calculations, relevant experiments should be proved to elucidate that water molecules tend to adsorb at the W site rather than the Ru site.*

Response to Comment 4: Thanks for your thoughtful suggestion.

According to relevant literature, the adsorption tendency of water molecules on catalysts can be compared by measuring the water contact angle (Adv. Mater. 2024, 36, 2308086; Adv. Funct. Mater. 2025, 35, 2415854; ACS Nano 2025, 19, 19, 18244-18255). The water contact angles of Ru/C catalyst and Ru/C+H_xWO₃ mixed catalyst are 41° and 16°, respectively, which indicates that the adding of H_xWO₃ into Ru/C can much reduce the contact angle, showing the better adsorption capability of water molecules on the W site than the case on the Ru sites (**Supplementary Fig. 32**).

The result was discussed in the main text at **Page 10 Paragraph 2** and **Supplementary Fig. 33**:

“...the Ru sites. In contrast, for Ru-H_xWO₃, H₂O molecules can spontaneously adsorb on both Ru and W sites, with the W sites having a more negative adsorption energy, which is confirmed by the contact angle (Supplementary Fig. 33). Unlike the W sites...”

Supplementary Fig. 33 Contact angles of water on **a** Ru/C and **b** Ru/C+H_xWO₃.

The water contact angles of Ru/C catalyst and Ru/C+H_xWO₃ mixed catalyst are 41° and 16°, respectively, which shows the better adsorption capability of water molecules on the W sites than the case on the Ru sites. **Note:** Ru/C+H_xWO₃ mixed catalyst was prepared by mixing finely ground H_xWO₃ powder with commercial Ru/C catalyst with a mass ratio of 1:1.

Comment 5. *Given the pivotal role of lattice hydrogen doping in theoretical modes, lattice hydrogen incorporation was quantitatively characterized via XRD should be performed to corroborate TGA-derived doping concentrations and ensure structure accuracy.*

Response to Comment 5: Thanks for your thoughtful suggestion.

In order to reveal the relationship between the lattice H concentration and the XRD diffraction angles, the WO_3 NN samples were thermal treated in H_2 atmosphere at different temperatures (200, 250, 300, 350 °C) for 60 min, denoted as H_xWO_3 NN-T. The XRD characterizations indicate that H_xWO_3 NN-200 keeps the (002) peak while H_xWO_3 NN-250, -300 and -350 show the two overlapped peaks corresponding to WO_3 and H_xWO_3 . This phenomenon could be resulted from the hydrogenation from surface to bulk. Therefore, it is difficult to get the samples to evaluate the relationship between lattice H concentration and the XRD diffraction angles.

Actually, according to the Response to Comments 2 of Reviewer 2, we quantified the x value in H_xWO_3 using electrochemical methods and TGA, and these results are consistent.

Fig. R6 XRD pattern for H_xWO_3 treated with different temperature in H_2 atmosphere.

Comment 6. *The current lattice spacing analysis (Figure S5) only examines two selected locations in Ru- WO_3 . To conclusively demonstrate that NaBH_4 reduction does not introduce lattice hydrogen into WO_3 , multiple spatially resolved measurements.*

Response to Comment 6: Thanks for your thoughtful suggestion.

The lattice spacing of the marked regions for WO_3 and Ru NPs was measured, which shows the interplanar distances of 0.205 nm for Ru NPs and 0.385 nm for WO_3 . This result confirms that the NaBH_4 treatment leads to the reduction of RuCl_3 to the metallic Ru and does not introduce lattice H into the WO_3 . In addition, the Ru- WO_3 NN after NaBH_4 treatment exhibits the unshifted (002) peak in XRD patterns (**Fig. 1e**), no obvious signal of $\delta(\text{WO-H})$ in Raman spectra (**Fig. 1f**) and no signals of bridging/terminal hydroxy groups in ^1H NMR spectra (**Fig. 1g**). These corroborating results clearly indicate that the NaBH_4 treatment does not introduce lattice hydrogen into the WO_3 NN.

The lattice spacing of the marked regions for Ru and WO_3 in HRTEM images was added into Supplementary Fig. 6:

Supplementary Fig. 6. Morphology and crystal structure of Ru-WO₃ NN. a-c SEM (a), TEM (b) and HRTEM (c) image of Ru-WO₃ NN. Inset in (c) is the local enlargement. d-e The lattice spacing corresponding to the marked region 1 for WO₃ (d) and region 2 for Ru NPs (e).

Comment 7. *Whether relative to WO₃ NN or H_xWO₃, the binding energy of Ru-H_xWO₃ moves towards higher binding energy rather than lower binding energy, which needs to be corrected.*

Response to Comment 7: Thanks for your thoughtful suggestion.

For the W 4f_{7/2} signal (**Fig. 2a**), the binding energy (BE) is 36.15 eV for WO₃ NN, 35.80 eV for H_xWO₃ NN and 35.65 eV for Ru-H_xWO₃ NN. It is clearly showed that, relative to WO₃ NN or H_xWO₃ NN, the BE of Ru-H_xWO₃ NN moves towards the lower BE side. It can be inferred that the negative shift of BE for W is due to the presence of the Ru-O-W interface, where the metallic state Ru gives electrons to O, forming relative positive valence state, and then, O further transfers electrons to the W element to form relative negative valence state.

Comment 8. *In EXAFS, the absence of Ru-W bonds cannot be excluded. A RuW alloy reference should be synthesized and measured to validate the above results.*

Response to Comment 8: Thanks for your thoughtful suggestion.

It is unfortunate that the pure RuW alloy is difficult to prepare and preserve. Based on the current XANES spectra for Ru and W, it is seen there are no obvious signals for W-W/Ru bond in Ru-H_xWO₃ NN and Ru-WO₃ NN (**Fig. 2e**). In addition, the XAFS spectra were calculated by DFT, as shown in **Supplementary Fig. 8**. To verify the validity of the calculation results, we first calculated the K-edge XANES spectrum and corresponding R-space EXAFS spectra for Ru of Ru foil. It can be found that the calculated results are basically consistent with the experimental results. Then, the Ru K-edge XANES spectrum for RuW alloy foil and corresponding R-space EXAFS spectra for Ru in Ru and RuW foils were calculated. It is seen the Ru K-edge XANES for RuW alloy is obviously different from those of Ru foil and Ru-H_xWO₃ NN. In the EXAFS spectra, the Ru signal for Ru-W foil shows an obvious widening and position shift to ~2.7 Å, much different from the Ru foil and Ru-H_xWO₃ NN. Therefore, the Ru NPs bond to the H_xWO₃ and WO₃ NN through Ru-O ionic bonds, rather than Ru-W metallic bonds.

Supplementary Fig. 8 XAFS spectra by calculation. a-b Model of Ru foil (a) and RuW foil (b). c Ru K-edge XANES spectra. d Corresponding k^2 -weighted R-space Fourier transformed EXAFS spectra for Ru.

The discussion was exchanged in main text at Page 5, Paragraph 1:

“...bonds, respectively. These results and corresponding theoretical calculations confirm that the Ru NPs bond to the H_xWO₃ and WO₃ NN through Ru-O ionic bonds, rather than Ru-W metallic bonds (Fig. 2f, and Supplementary Fig. 8).”

Comment 9. *In kinetic experiments, the selected potential should be noted in the calculation of the kinetic values in Figure S21e and f.*

Response to Comment 9: Thanks for your thoughtful suggestion.

The selected potential in kinetic experiments is -100 mV (vs. RHE), which was noted in Supplementary Fig. 24e and f.

Supplementary Fig. 24. KIE experiments. e,f Calculated KIE values (J_{H20}/J_{D20}) at 100 mV in 1 M PBS (e) and 1 M KOH (f).

Comment 10. *The calculation method of the normalized intensity value of the Raman spectra needs to be described in detail.*

Response to Comment 10: Thanks for your thoughtful suggestion. The Reviewer #1 also gave the same suggestion (see **Response to Comment 9 of Reviewer #1**).

The specific normalization method is performed as follows: using the peak intensity at 0 V (vs. RHE) as the reference value, and then calculate the ratio of the peak intensity at other potentials to the reference value.

The corresponding description was added in **Supplementary Fig. 25**:

“The normalization method: the peak intensity at 0 V (vs. RHE) was set as the reference value, and the normalized intensity at a certain potential is the ratio of the corresponding peak intensity to the reference value.”

Reviewer #4:

The manuscript by Zhang et al. proposes a novel concept based on the lattice hydrogen cycling mechanism for designing advanced electrocatalysts capable of achieving pH-universal hydrogen evolution reaction (HER) performance. Experimentally, this class of catalyst is realized by integrating a proton-blocking metal with a non-proton-blocking support. The resulting Ru-H_xWO₃ nanoneedle (NN) catalyst exhibits outstanding HER activity across a wide pH range, representing a state-of-the-art benchmark for pH-universal HER.

Achieving high HER performance at industrial current densities across varying pH conditions is crucial for the development of sustainable energy conversion technologies. This work addresses a key bottleneck and introduces a transformative strategy with the potential for broad applicability in renewable energy-powered processes. I recommend publication of the manuscript in Nature Communications. The following comments are provided to improve the manuscript:

General Response: Thanks for your encouraging comment.

Comment 1. *The Ru-WO₃ NN sample without lattice hydrogen was prepared via NaBH₄ treatment. However, NaBH₄ is typically considered a reducing agent that could introduce hydrogen into the lattice. Please clarify why NaBH₄ treatment in this case does not lead to lattice hydrogen insertion into WO₃.*

Response to Comment 1: Thanks for your thoughtful suggestion. The Reviewer #3 also gave the same suggestion (**Please see the Response to Comment 6 of Reviewer #3**).

The lattice spacing of the marked regions for WO₃ and Ru NPs was measured, which shows the interplanar distances of 0.205 nm for Ru NPs and 0.385 nm for WO₃. This result confirms that the NaBH₄ treatment leads to the reduction of RuCl₃ to the metallic Ru and does not introduce lattice H into the WO₃. In addition, the Ru-WO₃ NN after NaBH₄ treatment exhibits the unshifted (002) peak in XRD patterns (**Fig. 1e**), no obvious signal of δ(WO-H) in Raman spectra (**Fig. 1f**) and no signals of bridging/terminal hydroxy groups in ¹H NMR spectra (**Fig. 1g**). These corroborating results clearly indicate that the NaBH₄ treatment does not introduce lattice hydrogen into the WO₃ NN.

Comment 2. *The LSV curves presented in Figure 3 lack resolution in the low current density region. Please provide magnified plots of the low-current region to clearly demonstrate the intrinsic catalytic activity.*

Response to Comment 2: Thanks for your thoughtful suggestion.

The LSV curves in the low current density range (0~100 mA cm⁻²) were added as **Supplementary Fig. 9g-i** to demonstrate the intrinsic catalytic activity.

Supplementary Fig. 9. Polarization curves. g-i LSV curves with the low region of current density in 0.5 M H₂SO₄ (g), 1 M PBS (h) and 1 M KOH (i).

Comment 3. *The Tafel slopes of Ru-H_xWO₃ NN and the commercial Pt/C catalyst are reported to be approximately 30 mV dec⁻¹ in 0.5 M H₂SO₄ (Figure 3d and Supplementary Figure 14). However, deviations from this value are observed in 1 M PBS and 1 M KOH. Please provide a mechanistic explanation for these deviations.*

Response to Comment 3: Thanks for your thoughtful suggestion.

The Tafel slope reflects the kinetics of catalytic reactions. In neutral and alkaline environments, due to the obstruction of proton acquisition and transport, the Tafel slopes tend to be higher than that in acidic environment, which has also been reflected in other studies (Nat. Commun. 2021 12, 1369; Nat. Commun. 2024 15, 7475; Adv. Mater. 2022, 34, 2107548.).

The relevant description and Ref 10-12 have been added to **Supplementary Fig. 17:**

“In neutral and alkaline environments, due to the obstruction of proton acquisition and transport, the Tafel slopes tend to be higher than that in acidic environment¹⁰⁻¹².”

Comment 4. *The manuscript reports the mass activities of the catalysts but does not detail the method used for these calculations. Please include a description of the mass activity calculation procedure in an appropriate section of the manuscript.*

Response to Comment 4: Thanks for your thoughtful suggestion.

We have added the corresponding calculation method for mass activity in the section of **electrochemical testing methods** in Supplementary Materials at **Page 3 Paragraph 1:**

“...in different electrolytes. To get the activity of the catalyst per unit mass at the overpotential of 100 mV, the mass activities are calculated by the following formula: mass activities = $j_{100\text{mV}}/m(\text{Metal})$, where j represents the current density at the overpotential of 100 mV, m is the mass of precious metal in the catalyst.”

Comment 5. *The RRDE-based measurements of local pH reveal significant variations with applied potential, particularly in acidic and alkaline electrolytes. Please add a detailed explanation of the observed strong changes in local pH profiles under these conditions.*

Response to Comment 5: Thanks for your thoughtful suggestion.

As described in **Supplementary Materials** (Page 3), the OCP of the Pt-ring electrode ($E_{R,\text{ocp}}$) represents the equilibrium potential of the reaction $2\text{H}^+ + 2\text{e}^- \rightarrow \text{H}_2$, which varies with the local pH near the Pt-ring electrode (RE) according to the Nernst equation:

$$E_{R,ocp}(V \text{ vs. reference electrode}) = \frac{-2.303RT}{F} \text{pH}_R \quad (1)$$

The pH_D value of the catalyst-loaded disk electrode (DE) can be deduced from the pH_R value by the following equation:

$$c_{R,H^+} - c_{R,OH^-} = N_D(c_{D,H^+} - c_{D,OH^-}) + (1 - N_D)(c_{\infty,H^+} - c_{\infty,OH^-}) \quad (2)$$

where, N_D is the collection efficiency at the RE ($N_D = 0.37$), c_{∞,H^+} and c_{∞,OH^-} are the concentrations of H^+ and OH^- in the bulk electrolyte, respectively. It is seen that the $E_{R,ocp}$ can be used to deduce the pH_R (Equation 1) and further calculate the pH_D on the DE (Equation 2), and the c_R terms in Equation 2 are all obtained from the OCP testing. There is the specific $E_{R,ocp}$ corresponding to $\text{pH}_D=7$ during testing. Due to the significant impact of the last term in Equation 2, which is large in strong alkaline and acidic solutions, ultrasmall disturbances near the above specific $E_{R,ocp}$ can cause drastic changes of pH_D , which affects testing near this potential. However, this does not affect the trend over a large potential range.

A brief description about this point is provided in **Supplementary Fig. 20**:

“**Note:** It should be pointed out that this pH measurement method is more accurate under neutral conditions ¹. Due to the relatively large value of c_{∞,H^+} or c_{∞,OH^-} in Equation (2) in strongly acidic or alkaline environments, there is the specific OCP of Pt-RE corresponding to $\text{pH}=7$ on DE during testing. Ultrasmall disturbances near the above specific OCP of Pt-RE can cause drastic changes in pH on DE, but this does not affect the trend over a large potential range and the above conclusion was verified in this work.”

The list of changes

Manuscript				
Page	Line	Original version	Revised version	Note
3	15	Thermogravimetric analysis (TGA) further confirms the insertion of hydrogen into WO_3 NN, with x in range of 0.88~0.97 in H_xWO_3 NN (Supplementary Figs. 3).	CV tests and thermogravimetric analysis (TGA) further confirm the insertion of hydrogen into WO_3 NN, with x value of ~0.8 in H_xWO_3 NN (Supplementary Figs. 3 and 4).	Reviewer2 Comment2
3	35	-	...from 23.2° to 23.6° . The in situ XRD patterns show that the (002) peak gradually shifts to high angles during the temperature-increasing process, and the position of (002) peak keeps constant in the subsequent cooling process, which confirms the formation of H_xWO_3 during the thermal-hydrogenation (Supplementary Fig. 7) This shift....	Reviewer2 Comment1
5	10	These results confirm that the Ru NPs bond to the H_xWO_3 and WO_3 NN through Ru-O ionic bonds, rather than Ru-W metallic bonds (Fig. 2f).	These results and corresponding theoretical calculations confirm that the Ru NPs bond to the H_xWO_3 and WO_3 NN through Ru-O ionic bonds, rather than Ru-W metallic bonds (Fig. 2f, and Supplementary Fig. 8).	Reviewer3 Comment8

5	Fig. 2b			Reviewer2 Comment6
6	7 and 9	This electron redistribution causes the d -band centers of Ru and W in the Ru-H _x WO ₃ system to shift closer to the Fermi level (E_f) compared to those in Ru-WO ₃ , which can enhance the adsorption stability of key intermediates (e.g., H [*] , H ₂ O [*]) on Ru-H _x WO ₃ . Moreover, the higher Fermi level occupancy ($E - E_f = 0$) for Ru and W in Ru-H _x WO ₃ further facilitates the electron conductivity and HER thereof (Fig. 2h,i).	Add reference 23 and 24 This electron redistribution causes the d -band centers of Ru and W in the Ru-H _x WO ₃ system to shift closer to the Fermi level (E_f) compared to those in Ru-WO ₃ , which can enhance the adsorption stability of key intermediates (e.g., H [*] , H ₂ O [*]) on Ru-H _x WO ₃ ²³ . Moreover, the higher Fermi level occupancy ($E - E_f = 0$) for Ru and W in Ru-H _x WO ₃ further facilitates the electron conductivity and HER thereof (Fig. 2h,i) ²⁴ .	Reviewer1 Comment4
10	3	Moreover, while the ΔG_{H^*} for the top Ru sites remains highly positive ($\Delta G_{H^*} = 0.50$ eV), the ΔG_{H^*} for the interfacial Ru sites turns slightly negative, with a near-zero value of -0.08 eV, which is ideal for HER (Fig. 5b,c) ³⁴ .	Moreover, the ΔG_{H^*} for the Ru _{Interfacial} in H _x WO ₃ is -0.08 eV, very close to the ideal value for HER. In contrast, the ΔG_{H^*} for the possible Ru and W active sites remain highly positive, showing the inferior HER activity of these sites (Fig. 5b,c, Supplementary Fig. 32) ³⁴ .	Reviewer1 Comment6 and 7
10	Fig. 5c			Reviewer1 Comment6 and 7
10	21	To understand this phenomenon, we calculated the water dissociation process on each metal site in Ru-WO ₃ and Ru-H _x WO ₃	To understand this phenomenon, we calculated the water dissociation process on each metal site in Ru-WO ₃ and Ru-H _x WO ₃ (Supplementary Fig. 31).	Reviewer3 Comment1
10	24	In contrast, for Ru-H _x WO ₃ , H ₂ O molecules can spontaneously adsorb on both Ru and W sites, with the W sites having a more negative adsorption energy.	In contrast, for Ru-H _x WO ₃ , H ₂ O molecules can spontaneously adsorb on both Ru and W sites, with the W sites having a more negative adsorption energy, which is confirmed by the contact angle (Supplementary Fig. 33).	Reviewer3 Comment4
Supplementary Materials				
Page	Line	Original version	Revised version	Note
2	11	-	Ir/C (5 wt%) was purchased from Meryer Chemical Technology Co.,Ltd.	Reviewer1 Comment10
3	8	-	To get the activity of the catalyst per unit mass at the overpotential of 100 mV, the mass activities are calculated by the formula mass	Reviewer4 Comment4

			activities = $j_{100\text{mV}}/m(\text{Metal})$, where j represents the current density at the overpotential of 100 mV and m represents the mass of the precious metal .	
7	1	-	Add Supplementary Fig. 3. CV curves, deduced Q_a and x value in 0.5 M H_2SO_4, 1 M PBS and 1 M KOH. a-f CV curves. g,h deduced Q_a and x value. The CV tests were used to quantify the amount of reversible hydrogen insertion in WO_3 NN and H_xWO_3 NN⁹. The WO_3 NN and H_xWO_3 NN powders were stripped from the Cu foam by ultrasonication, and then coated on carbon paper as working electrode with a mass loading of 1 mg cm^{-2}. The CV scan rate is 10 mV s^{-1}. The integral of the positive current with respect to time for one CV cycle is taken to calculate the Q_a, which was converted to the number of hydron atom per tungsten atom by the following equations: $Q_a = \int j dt$ $n_H = \frac{Q_a \times N_e}{N}$ $x = \frac{n_H}{n_{\text{WO}_3}}$ where j is the positive current, n_H is the amount of substance for H insertion, N_e is the number of electrons per Coulomb; N is the Avogadro constant; n_{WO_3} is the amount of substance for WO_3. It was found that after the H insertion via electrochemical method, the x value of WO_3 NN was 0.45 in 0.5 M H_2SO_4, similar with the above report; and the x value of H_xWO_3 NN was 0.80 in 0.5 M H_2SO_4 which was obviously higher than that of WO_3 NN. For H_xWO_3 NN, the measured x value was the amount of H that can achieve reversible hydrogen insertion. In addition, the x values of WO_3 NN were much lower when tested in neutral ($x=0.18$) and alkaline ($x=0.26$) electrolytes due to the increased difficulty in obtaining H species, indicating a poor hydrogen supply for WO_3 NN during HER. However, the x values of H_xWO_3 NN were 0.6 in neutral electrolyte and 0.68 in alkaline electrolyte, thanks to its high initial concentration and faster replenishment of lattice hydrogen, which is contributed to achieving similar HER performance in	Reviewer2 Comment2

			different electrolyte.	
8	1	-	Add It is known that there is no change in the amount of substance (n) for WO_3 during TGA. Therefore, the value of x can be calculated using the following formula: $m_{\text{H}} = m_{\text{WO}_3-\text{H}_2} - m_{\text{WO}_3-\text{Ar}} = \Delta\omega \times m_{\text{initial}}$ $x \times n \times M_{\text{H}} = \Delta\omega \times n \times M_{\text{WO}_3}$ $x = \frac{M_{\text{WO}_3} \times \Delta\omega}{M_{\text{H}}}$ where M is the molecular weight, $\Delta\omega$ is the weight-loss difference, and m is the mass.	Reviewer2 Comment3
10	1	-	Add Supplementary Fig. 6d,e. d-e The lattice spacing corresponding to the marked region 1 for WO_3 (d) and region 2 for Ru NPs (e).	Reviewer3 Comment6
11	1	-	Add Supplementary Fig. 7. In situ XRD patterns for WO_3 sample thermal-hydrogenated at different temperatures under 10% H_2/Ar atmosphere. The local enlargements of the patterns in the range of 20°-30° are shown for clarity. The WO_3 NN arrays on Cu foam was thermal-hydrogenated in 10% H_2/Ar atmosphere and the XRD patterns were in situ recorded on Smartlab SE with a temperature interval of 50°C in the range of 100°C to 400°C.	Reviewer2 Comment1
12	1	-	Add Supplementary Fig. 8. XAFS spectra by calculation. a,b Model of Ru foil (a) and RuW foil (b). c Ru K-edge XANES spectra. d Corresponding k^2-weighted R-space Fourier transformed EXAFS spectra for Ru. To verify the validity of the calculation results, we first calculated the K-edge XANES spectrum and corresponding R-space EXAFS spectra for Ru of Ru foil. It can be found that the calculated results are basically consistent with the experimental results. Then, the Ru K-edge XANES spectrum for RuW alloy foil and corresponding R-space EXAFS spectra for Ru in Ru and RuW foils were calculated. It is seen the Ru K-edge XANES for RuW alloy is obviously different from those of Ru foil and $\text{Ru-H}_x\text{WO}_3$ NN. In the EXAFS spectra, the Ru signal for Ru-W foil shows an obvious widening and position shift to $\sim 2.7 \text{ \AA}$, much	Reviewer3 Comment8

			different from the Ru foil and Ru-H _x WO ₃ NN. Therefore, the Ru NPs bond to the H _x WO ₃ and WO ₃ NN through Ru-O ionic bonds, rather than Ru-W metallic bonds.	
13	1	-	Add Supplementary Fig. 9d-i . Polarization curves. d-f In 0.5 M H ₂ SO ₄ (a), 1 M PBS (b) and 1 M KOH (c) without iR compensation. g-i In 0.5 M H ₂ SO ₄ (a), 1 M PBS (b) and 1 M KOH (c) with low current range.	Reviewer1 Comment8 and Reviewer4 Comment2
16	1		Exchange Supplementary Fig. 12 .	Reviewer2 Comment5
21	2	-	In neutral and alkaline environments, due to the obstruction of proton acquisition and transport, the Tafel slopes tend to be higher than that in acidic environment ¹⁰⁻¹²	Reviewer4 Comment3
24	7	-	Note: It should be pointed out that this pH measurement method is more accurate under neutral conditions. Due to the relatively large value of c_{∞, H^+} or c_{∞, OH^-} in Equation (2) in strongly acidic or alkaline environments, there is the specific OCP of Pt-RE corresponding to pH=7 on DE during testing. Ultrasmall disturbances near the above specific OCP of Pt-RE can cause drastic changes in pH on DE, but this does not affect the trend over a large potential range and the above conclusion was verified in this work.	Reviewer1 Comment5 and Reviewer4 Comment5
28	1		Exchange Supplementary Fig. 24d,f .	Reviewer3 Comment9
29	7	-	The method of normalization: the peak intensity of 0 V vs. RHE was set as the reference peak intensity, and then calculate the ratio of the peak intensity at other potentials to the reference peak intensity.	Reviewer1 Comment9 and Reviewer3 Comment10

34	1		Exchange Supplementary Fig. 30.	Reviewer1 Comment3
35	1	-	Add Supplementary Fig. 31d d The process of H₂O molecule dissociation on W sites in Ru-H_xWO₃.	Reviewer3 Comment1
36	1	-	Add Supplementary Fig. 32 Calculation for H adsorbed on W sites. a WO₃. b H_xWO₃. c Corresponding free energy profiles for HER on single W sites. d Corresponding free energy profiles for HER on adjacent W sites.	Reviewer1 Comment6 and 7
37	1	-	Add Supplementary Fig. 33 Supplementary Fig. 33 Contact angles of water on (a) Ru/C and (b) Ru/C+H_xWO₃. The water contact angles of Ru/C catalyst and Ru/C+H_xWO₃ mixed catalyst are 41° and 16°, respectively, which shows the better adsorption capability of water molecules on the W sites. Note: Ru/C+H_xWO₃ mixed catalyst was prepared by mixing finely ground H_xWO₃ powder with commercial Ru/C catalyst with a mass ratio of 1:1.	Reviewer3 Comment4
38	1		Exchange Supplementary Fig. 34	Reviewer1 Comment10

The sequence numbers of Figures, References in the revised manuscript and Supplementary Materials are changed accordingly.

The point-by-point response to the reviewers' comments

(For clarity, the reviewers' comments are quoted in *italics* ahead of the corresponding responses.)

Reviewer #1:

The revised manuscript has been significantly improved and is now ready for publication in NC.

General Response: We truly appreciate your kind help to improve this paper.

Reviewer #2:

The revised manuscript is acceptable.

General Response: We truly appreciate your kind help to improve this paper.

Reviewer #3:

In this manuscript, the authors presented a novel $Ru-H_xWO_3$ NN HER catalyst a lattice-hydrogen cycling mechanism that dissociates hydrogen intermediate availability from electrolyte pH. Although the authors have revised the manuscript, several critical issues remain, which should be addressed before publication. The detailed comments are listed as follows.

General Response: Thanks for your encouraging comments and constructive suggestions.

Comment 1. *The synthesis method of electrocatalysts in this work is the same as that in the literature (DOI: 10.1038/s41467-022-33007-3), but the explanation is somewhat different. The XRD shift to higher angles is due to the introduction of oxygen vacancies rather than lattice hydrogen. The insertion of lattice hydrogen usually causes lattice expansion, which should lead to a shift towards a lower angle in the XRD pattern. This is consistent with the conclusion stated in the literature (DOI: 10.1038/s41467-023-39963-8). Moreover, the calculation of the amount of inserted hydrogen by the electrochemical method is not accurate. Since H_xWO_3 already contains lattice hydrogen, the amount of electrochemically inserted hydrogen should be lower. So, related discussion should be provided.*

Response to Comment 1: Thanks for your meaningful questions. We will respond your question one by one.

- (1) **About the synthesis method.** In the literature (DOI: 10.1038/s41467-022-33007-3), the product was prepared on carbon paper, while the product was synthesized on the Cu foam in this work, followed by the similar thermal treatment in H_2 . We found that the Cu foam is critical to the H insertion of the WO_3 NN arrays. According to the literatures (Dopov. Akad. Nauk Ukr. RSR, Ser. B, 1985, 1, 46-49; J. Mater. Sci., 1988, 23, 267; J. Alloys Compd., 2001, 320, 1-6), the presence of Cu elements could boost the H insertion into WO_3 . If we replaced the Cu foam substrate with carbon paper (CP) or carbon cloth (CC), no significant shifts were observed for the (002) peak after the H_2 treatment for the WO_3 nanoarrays deposited on CP and CC ($WO_3@CP$, $WO_3@CC$) (**Figure R1a**), different from the samples grown on the Cu foam. Therefore, the existence of Cu foam is beneficial for the H insertion of the WO_3 NN arrays.
- (2) **About the XRD shift.** The XRD shift of the H_xWO_3 NN to higher angles should be ascribed to the introduction of lattice hydrogen. Firstly, the Raman spectra and 1H NMR spectra have clearly proved the presence of lattice H in the H_xWO_3 NN, consistent with the XRD shift (**Figure 1f,g**). The DFT calculation shows that the insertion of lattice hydrogen causes lattice contraction (**Figure R2**), rather than lattice expansion, consistent with the standard XRD patterns for H_xWO_3 and WO_3 (**Figure R3**). In addition, the XPS analysis indicates that the oxygen vacancy concentrations have no obvious difference for the WO_3 NN and H_xWO_3 NN (**Figure R1b**). These results indicate that, in this paper, the XRD shift of the H_xWO_3 NN relative to the WO_3 NN is due to the H insertion rather than the formation of oxygen vacancies, which

could be stemmed from the promoted H insertion by the Cu substrate. Worthy to mention is that, in the suggested literature (DOI: 10.1038/s41467-023-39963-8), the XRD peaks of H_xWO_3 and Ir- H_xWO_3 catalysts with oxygen vacancies are shifted toward a low-angle direction, rather than toward a high-angle direction shown in **Figure 1e**.

Figure R1 (a) XRD patterns of WO_3 on CP and CC before and after H_2 calculation. (b) O 1s XPS spectra of WO_3 NN and H_xWO_3 NN.

- (3) **About the calculation of the inserted hydrogen amount.** The calculation of the inserted hydrogen amount by the electrochemical method is described in **Response to Comment 2**. Actually, the measured x value refers to the amount of H that can be reversibly inserted rather than extra H that can be inserted. Because the reversibly insertable H includes both thermally inserted H and electrochemically inserted H, H_xWO_3 contains more H amount while WO_3 only contains electrochemically inserted H, therefore the x value of H_xWO_3 is higher than that of WO_3 .

Figure R2. Lattice spacings of (a) WO_3 and (b) H_xWO_3 by DFT calculation.

Figure R3. Standard JCPDF XRD patterns for WO_3 and H_xWO_3 .

Comment 2. Regarding the calculation of the amount of inserted hydrogen by the electrochemical method, there are a few questions.

a. Q_a should be the integral of current over time. If current density is used, it should be multiplied by the electrode area.

b. According to the supporting information in reference 9, this method is not accurate for quantification, as it takes into account the current generated by the hydrogen evolution reaction, leading to a value lower than the actual one.

c. In supplementary Fig. 3h, why is the calculated hydrogen content in H_xWO_3 higher than that in WO_3 ? Since H_xWO_3 already contains lattice hydrogen, shouldn't the amount of electrochemically inserted hydrogen be lower?

Response to Comment 2: Thanks for your valuable question.

a. The electrode area is 1 cm^2 , and the mass loading is 1 mg cm^{-2} ; therefore, the absolute value of current is equal to current density.

b. In reference 9, a wide potential range of $-0.5\sim-0.6\text{ V}$ (vs. RHE) was used for calculating the amount of inserted hydrogen. Hence, some HER current at the lower potential range could be included to disturb the calculation of x value. In this work, we chose a narrow potential range of $-0.2\sim-0.6\text{ V}$ (vs. RHE) to reduce the impact of HER current. In addition, the carbon paper used for CV test, which is mentioned in the Method 3 in Supporting Information of reference 9, can coat relatively more catalysts compared to thin film electrodes on glassy carbon which, thereby, can effectively reduce the impact of HER current ($Q_{\text{HER}} \ll Q_{\text{insert}}$). Therefore, we think the value of the amount of inserted hydrogen calculated by this method is relatively accurate.

c. The measured x value refers to the amount of H that can be reversibly inserted rather than extra H that can be inserted. Because the reversibly insertable H includes both thermally inserted H and electrochemically inserted H, H_xWO_3 contains more H amount while WO_3 only contains electrochemically inserted H, therefore the x value of H_xWO_3 is higher than that of WO_3 .

Comment 3. For DFT calculations, how is the H_xWO_3 model constructed, and how is the hydrogen content (x) determined? Based on the TGA results, the x value in H_xWO_3 NN support is about 0.88-0.97. However, in supplementary Fig. 30, is the x value of the H_xWO_3 model 0.75? In comment 3 for reviewer 1, the x for the Ru- H_xWO_3 model is 0.625. What is the basis for this value?

Response to Comment 3: Thanks for your thoughtful question.

We are sorry for the numerical labeling errors for H_xWO_3 and Ru- H_xWO_3 models in Supplementary Fig. 30. The H_xWO_3 model is $H_{22}W_{36}O_{108}$, which has the x value of 0.611 (Supplementary Fig. 30).

Supplementary Fig. 30. The side illustrations of models for DFT calculation. **a** WO_3 . **b** H_xWO_3 . **c** $Ru-WO_3$. **d** $Ru-H_xWO_3$.

The above theoretical models were constructed for DFT calculation based on the XRD patterns (XRD standard card PDF: # 84-2460 in **Fig. 1e**) and EXAFS results (**Fig. 2c-f**). The relevant coordination information was added in Supplementary Fig. 30. For the convenience of calculation in all-pH conditions, the amount of H per single W was set as ~ 0.625 , i.e., $H_{22}W_{36}O_{108}$, which was referenced by the literature (J. Am. Chem. Soc. 2022, 144, 6420–6433.) with the x of 0.625 and the calculated x values in neutral (0.6)/alkaline (0.68) electrolyte by electrochemical insertion (Supplementary Fig. 3). In the HER and hydrogen migration process, an extra H atom was added in the H_xWO_3 for calculation, and the model was set as $H_{23}W_{36}O_{108}$. Therefore, on average, the amount of H per single W was set as ~ 0.625 .

Comment 4. *The synthesis method used in this work is the same as that in the literature (DOI: 10.1038/s41467-022-33007-3), but the explanation for the XRD peak shift toward higher angles is completely different. The reference attributes it to the presence of oxygen vacancies, rather than hydrogen insertion. According to another literature (DOI: 10.1038/s41467-023-39963-8), the insertion of hydrogen leads to a shift of the XRD diffraction peaks of H_xWO_3 toward lower angles in supplementary Fig. 10. How should this be understood?*

Response to Comment 4: Thanks for your valuable questions.

Please see the **Response to Comment 1**.

Comment 5. *According to the literature (DOI: 10.1038/s41467-022-33007-3), with an almost identical synthesis method, the oxygen vacancies and reversible hydrogen spillover play a crucial role in enhancing the HER activity. This work did not take into account the influence of oxygen vacancies.*

Response to Comment 5: Thanks for your valuable question.

Please see the **Response to Comment 1**.

Comment 6. In supplementary Fig. 21, the HER performance of Ru- H_x WO₃ is better than that of Ru-WO₃, which cannot be simply attributed to the effect of lattice hydrogen. The two materials were synthesized using completely different methods: the former was reduced by H₂ annealing, while the latter was reduced by NaBH₄. It may introduce the B element.

Response to Comment 6: Thanks for your valuable question.

The presence of B element in WO₃ NN and Ru-WO₃ NN treated with NaBH₄ was analyzed by XPS. It can be observed that no B signals were detected in B 1s fine spectra (**Figure R4**). Therefore, the influence of B element on the HER performance can be excluded.

Figure R4 B 1s XPS spectra of WO₃ NN and Ru-WO₃ NN treated with NaBH₄.

Comment 7. In supplementary Fig. 20c, the pH value in the KOH solution changed from 13 to 2, which is indeed a bit strange.

Response to Comment 7: Thanks for your valuable question.

As proposed in literatures (Nat. Commun. 2023, 14, 4209; Chem. Sci. 2024, 15, 5385-5402.), the change of surface lattice hydrogen in tungsten oxide with the H⁺ concentration is a dynamic and reversible process. Therefore, due to the presence of high concentrations of surface lattice hydrogen, H_xWO₃ NN can exhibit low pH on the surface and even in the solution near the surface.

Comment 8. In Figure 4d-f, the decrease in the Raman signals of O-W-O indicates the insertion of hydrogen. Meanwhile, the signals of W-O-H should increase. Only H_xWO₃ and Ru-WO₃ conform to this phenomenon. However, for Ru-H_xWO₃, the Raman signal of WO-H decreases, which is quite strange.

Response to Comment 8: Thanks for your valuable question.

Actually, only H_xWO₃ conforms the phenomenon of the decrease in the Raman signal of O-W-O and the increase of W-O-H signal. For the Ru-WO₃, this phenomenon was observed in the low potential range, i.e., OCP~-0.05 V to firstly insert lattice hydrogen in WO₃, while the signal of W-O-H is very weak. In the high potential range, it can be ascribed to the H migration from W-O-H to the Ru sites for HER, achieving a fast lattice-hydrogen cycling during HER in Ru-WO₃ and Ru-H_xWO₃. The specific process is that the lattice hydrogen in H_xWO₃/WO₃ can efficiently supply H species to the Ru sites via the lattice-H migration. The consumed lattice-H is spontaneously replenished in all-pH electrolytes. Because of the efficient lattice-H cycling, when the consumed

lattice-H rate is higher than the replenished lattice-H, the decreases of WO-H signals occurred. For H_xWO_3 , the inferior HER performance consumes less lattice-H, thus the signals of W-O-H increase.

Reviewer #4 (Remarks to the Author):

The manuscript can be accepted in the present form.

Response: We truly appreciate your kind help to improve this paper.

The list of changes

Supplementary Materials				
Page	Line	Original version	Revised version	Note
34	1	Original version of Figure 30 showing four panels (a, b, c, d) illustrating crystal structures. Panel (a) shows $W_{36}O_{108}$, (b) shows $H_{27}W_{36}O_{108}$, (c) shows $Ru_{19}W_{36}O_{108}$, and (d) shows $Ru_{19}H_{27}W_{36}O_{108}$.	Exchange Supplementary Fig. 30. Revised version of Figure 30 showing four panels (a, b, c, d) illustrating crystal structures. Panel (a) shows $W_{36}O_{108}$, (b) shows $H_{27}W_{36}O_{108}$, (c) shows $Ru_{19}W_{36}O_{108}$, and (d) shows $Ru_{19}H_{27}W_{36}O_{108}$.	Reviewer3 Comment3
The sequence numbers of Figures, References in the revised manuscript and Supplementary Materials are changed accordingly.				

Remarks to the Author: **Minor revision**

This study proposes an innovative lattice hydrogen cycling concept, which decouples H^{*} availability from electrolyte pH while mitigating local pH fluctuations, the critical limitation in pH-universal HER. The catalyst was experimentally designed through integrating the proton-blocking metallic components with non-proton-blocking support materials (Ru-H_xWO₃ NN catalyst). Consistently, due to the lattice-hydrogen cycling pathway, Ru-H_xWO₃ NN exhibited unprecedented pH-universal performance. It is thus expected that the proposed strategy and associated mechanistic insights will guide the rational design of advanced catalysts in various hydrogen-related electrochemical processes. Overall, the manuscript presents a logically structured investigation and merits consideration for publication after addressing the following concerns.

1. In this study, the samples as-synthesized exhibit nanoneedle morphology. However, nanostructures with sharp geometrical features typically demonstrate tip-enhanced effects, so does this effect affect the performance of the catalyst in this work?
2. Metallic substrates tend to have a greater influence on the catalytic reaction. In order to exclude the influence of the substrate, could the copper foam used in the manuscript be replaced with another substrate, such as carbon paper?
3. How are WO₃, H_xWO₃, Ru-WO₃ and Ru-H_xWO₃ modelled? Do their configurations, ratios, and coordination numbers consistent with experiment?
4. The authors posited that variations in the d-band center position and Fermi-level occupied states significantly influence HER process. Please add some representative references after this conclusion.
5. In Supplementary Fig. 17a and c, a significant pH mutation (from ~2 to ~12) is observed. This may not be entirely reasonable, and the authors should provide a brief explanation for this phenomenon.
6. The authors considered Ru and O as active sites. Although W sites in WO₃ was not suitable for HER, the author should add corresponding calculation of W sites for comparison to maintain rigor.
7. In Fig. 5c, only two Ru sites were listed, which cannot reflect superior HER activity of interfacial Ru compared to other sites. The authors should add comparisons with other active sites to enhance persuasiveness.
8. The LSV data for HER without iR compensation needs to be added;
9. In Supplementary Fig. 22, the authors implemented intensity normalization of relative peaks to facilitate their observation. However, the current manuscript lacks corresponding description regarding the specific normalization methodology employed. It is recommended

that the authors incorporate a concise description to prevent potential ambiguities.

- 10.** The authors proved the general applicability of the proposed strategy. To further strengthen the comparative analysis, the authors should measure LSV curves of commercially available benchmark counterparts like Ir/C and Pt/C catalysts.